# Bacillus subtilis biofilm matrix components target seed oil bodies to promote growth and anti-fungal resistance in melon

M. V. Berlanga-Clavero [1], C. Molina-Santiago [1], A. M. Caraballo-Rodríguez [2], D. Petras [2,3], L. Díaz-Martínez[1], A. Pérez-García[1], A. de Vicente [1], V. J. Carrión[4,5], P. C. Dorrestein [2,6] and D. Romero [1]✉

**Beneficial microorganisms are used to stimulate the germination of seeds; however, their growth-promoting mechanisms remain largely unexplored. *Bacillus subtilis* is commonly found in association with different plant organs, providing protection against pathogens or stimulating plant growth. We report that application of *B. subtilis* to melon seeds results in genetic and physiological responses in seeds that alter the metabolic and developmental status in 5-d and 1-month-old plants upon germination. We analysed mutants in different components of the extracellular matrix of *B. subtilis* biofilms in interaction with seeds and found cooperation in bacterial colonization of seed storage tissues and growth promotion. Combining confocal microscopy with fluorogenic probes, we found that two specific components of the extracellular matrix, amyloid protein TasA and fengycin, differentially increased the concentrations of reactive oxygen species inside seeds. Further, using electron and fluorescence microscopy and metabolomics, we showed that both TasA and fengycin targeted the oil bodies in the seed endosperm, resulting in specific changes in lipid metabolism and accumulation of glutathione-related molecules. In turn, this results in two different plant growth developmental programmes: TasA and fengycin stimulate the development of radicles, and fengycin alone stimulate the growth of adult plants and resistance in the phylloplane to the fungus *Botrytis cinerea*. Understanding mechanisms of bacterial growth promotion will enable the design of bespoke growth promotion strains.**

S eed germination is a complex biological process that is controlled by interconnected hormone-regulated pathways[1]. Seed inoculation with plant growth-promoting rhizobacteria is commonly used to enable bacterial colonization of plants and beneficial effects of plant growth-promoting rhizobacteria post germination[2]. A detailed understanding of the mechanisms that underpin growth-promoting host–microbe interactions is important to enable design of innovative biotechnological probiotics.

*Bacillus subtilis* and closely related species coexist with plants, providing multiple beneficial services[3]. Production of polyvalent secondary metabolites, sporulation or biofilm formation underpin bacterial fitness and bioactivity towards plants[4]. Biofilms are formed by bacterial cells that are embedded in a secreted extracellular matrix (ECM)[5] that is essential for efficient colonization of plant organs[6]. The ECM also comprises secondary metabolites that mediate bacterial communication with the plants by triggering physiological responses associated with defence or growth[7].

Here we investigated the mechanism by which melon seeds respond to the stimulatory activity of *B. subtilis* and evaluated the functions of the ECM in this microbe–host interaction.

## Results

**B. subtilis regulates metabolism and growth of colonized plants.** Beneficial bacteria can promote two different and genetically controlled stages of seed germination, namely germination itself and growth of the emergent radicle[8]. Melon seeds treated with *B. subtilis* NCIB 3610 produced larger radicles compared with those grown from untreated seeds (Fig. 1a). However, the germination rates (initial emergence of the radicle) of treated seeds did not change compared with untreated seeds. We analysed the expression levels of *GA20ox1* and *CYP07A1*, which are two genes involved in the modulation of the determinant ratio of abscisic acid and gibberellins[9], and found that they are statistically similar between treated and untreated seeds (Extended Data Fig. 1a,b). Analysis of transcriptomes of three pools of seeds 16 h after treatment with a suspension of *B. subtilis* did not identify any changes in the expression levels of the genes involved in the germination-related hormone signalling pathway. However, the upregulation of plant genes involved in carbon metabolism and photosynthesis, together with repression of plant heat shock proteins and structural proteins of lipid storage vesicles (oleosin and caleosin), was detected and enabled activation of seed metabolism (Fig. 1b and Extended Data Fig. 1c,d)[10–12].

Metabolomic profiles of treated and untreated seeds were compared to reveal increased abundance of energetic resources, such as glycerophospholipids and fatty acyls with analogy to triacylglycerides (TAGs) in the initial stages (0 h) after treatment with *B. subtilis* cells (Fig. 1c and Extended Data Fig. 2a). At 24 h after the treatment, metabolite levels were similar between both conditions,

[1]Departamento de Microbiología, Instituto de Hortofruticultura Subtropical y Mediterránea 'La Mayora', Universidad de Málaga-Consejo Superior de Investigaciones Científicas (IHSM-UMA-CSIC), Universidad de Málaga, Málaga, Spain. [2]Collaborative Mass Spectrometry Innovation Center, Skaggs School of Pharmacy and Pharmaceutical Sciences, University of California San Diego, La Jolla, CA, USA. [3]CMFI Cluster of Excellence, Interfaculty Institute of Microbiology and Medicine, University of Tuebingen, Tuebingen, Germany. [4]Institute of Biology, Leiden University, Leiden, the Netherlands. [5]Department of Microbial Ecology, Netherlands Institute of Ecology (NIOO-KNAW), Wageningen, the Netherlands. [6]Center for Microbiome Innovation, University of California at San Diego, La Jolla, CA, USA. ✉e-mail: diego_romero@uma.es

although specific changes in the abundance of metabolites within these lipidic classes remained clear (Extended Data Fig. 2a).

These post-germination stimulatory effects of treatment with *B. subtilis* were long-lasting, considering that 1-month-old adult plants emerging from treated seeds developed a more vigorous radicular system and canopy than those emerging from untreated seeds (Fig. 1c). We analysed the metabolome of three adult plants that emerged from treated seeds to define putative metabolic changes that were associated with this long-lasting growth-promoting effect. Carboxylic acids, lipids and lipid-like molecules and organooxygen compounds were the main classes of metabolites differentially detected in aerial regions (leaves and stem) of plants grown from treated and untreated seeds (Extended Data Fig. 2b), while metabolomic composition of the roots was not significantly different between treated and untreated groups.

Changes in fatty acyls, carboxylic acids, organooxygen compounds and prenol lipids were the main metabolic signatures of the leaves from plants grown after bacterial treatment of the seeds (Fig. 1e). Our metabolomic analysis also revealed differential accumulation of L-tryptophan and cinnamic acids in leaves of adult plants derived from treated seeds (Extended Data Fig. 2c). Both families of molecules are biomarkers of beneficial plant–bacteria interactions[13,14].

### *B. subtilis* extracellular matrix and growth promotion.

We investigated the growth dynamics of the *B. subtilis* population during the first 5 d after seed treatment in the seed and emergent radicle (Fig. 2a) by colony-forming unit (c.f.u.) plating over seed and radicle extracts. In the radicles, the population remained unchanged and almost every cell sporulated upon radicle emergence (2 d after treatment). In the seeds, however, the bacterial population increased during the first 5 d post treatment and became almost entirely sporulated 4 d after bacterization. These results suggest that *B. subtilis* cells, mostly spores, may have been passively dragged by the emergent radicle or might have colonized and proliferated inside seeds. We used scanning electron microscopy analysis (SEM) of treated seeds to confirm that *B. subtilis* colonizes the inner regions of the seeds (Extended Data Fig. 3a). Confocal laser scanning microscopy (CLSM) analyses of transversely sectioned seeds previously treated with fluorescently labelled *B. subtilis* (CellTracker cm-DII) indicated that bacterial cells accumulate in the storage tissues near the seed micropyle, which is a natural entry point for bacteria into seeds[15] (Extended Data Fig. 3b). The cell densities of the wild-type (WT) strain and Δ*hag* (flagellum), Δ*eps* or Δ*srf* mutant strains, which are known to have altered swimming, sliding or swarming motility, respectively[16–18], were unchanged in two differentiated parts inside the seeds (Fig. 2b,c). Overall, these findings reveal that

growth-promoting activity might be triggered by bacterial cells that enter and colonize the seed storage tissues, which is a process that does not appear to rely specifically on any one type of bacterial motility.

Several studies have reported multifaceted contributions of bacterial biofilms to interactions with hosts[6,19–21]. The cell densities of single structural mutants Δ*tasA*, Δ*tapA*, Δ*eps* and Δ*bslA*, which form altered biofilms, were substantially decreased in seed extracts compared with that of the wild-type bacteria (set at 100%) 5 d after seed treatment (Fig. 2d). However, this result was not associated with the attachment of bacterial cells to the seeds a few hours after treatments (Extended Data Fig. 3c).

A similar bacterial population pattern was monitored in the emergent radicles, which, as described above for WT, may reflect dragging along of bacterial cells (Extended Data Fig. 3d). The Δ*srf*, Δ*pks* or Δ*pps* strains, which do not produce the secondary metabolites surfactin, bacillaene or fengycin, respectively (also present in the ECM), had unchanged population dynamics compared with the WT strain (Fig. 2d and Extended Data Fig. 3d). The decrease in the bacterial cell population size found in the structural mutants suggests that mutations led to a significant decrease in radicle growth promotion compared with treatment with the wild-type strain (Fig. 2e). In contrast to a correlation between bacterial cell density and plant growth promotion, the poorly persisting Δ*tasA* strain retained promoting activity, while the Δ*pks* and Δ*pps* strains failed to promote radicle growth despite considerable persistence on the seeds (Fig. 2d,e).

The absence of the amyloid protein TasA has recently been reported to provoke important cellular and physiological changes in *B. subtilis*[6]. To clarify whether the role of TasA in the structure of the ECM is related to TasA-dependent growth-promoting activity, we used *B. subtilis* strain JC81, which expresses a version of TasA that fails to restore biofilm formation but reverts the physiological status of Δ*tasA* cells to wild-type levels[6]. Treatment of seeds with JC81 failed to promote radicle growth, but the bacteria persisted on the seeds at a level comparable to that of the WT, Δ*pks* or Δ*pps* strains (Fig. 2d,e).

These results led us to propose three distinct and complementary contributions of ECM components to the promotion of seed radicle growth: first, EPS, BslA and TapA have a role in the persistence of *B. subtilis* cells in the seeds; second, fengycin (*pps*) and bacillaene (*pks*) have roles in the chemical dialogue of *Bacillus* with seeds; and third amyloid TasA is involved in both of these functions.

Comparison of the metabolic status of radicles 5 d after treatment of seeds with WT or various ECM mutant strains demonstrated a specific metabolic response of plants to chemicals present in the *B. subtilis* ECM (Extended Data Fig. 4a). The changes associated with bacterial treatment of seeds included a decrease in organooxygen

**Fig. 1 | Interaction of *B. subtilis* with the seeds stimulates radicle development and results in growth-promoting effect on adult plants. a**, Left: average ± s.d. radicle areas after seed treatments with *B. subtilis*. Statistical significance was assessed by a two-tailed *t*-test (*n* = 8; *P* = 0.0056). Right: representative radicles from untreated seed (left) and a *B. subtilis*-treated seed (right) 5 d after treatments. **b**, Volcano plot of DEGs identified by RNA-seq in bacterized seeds and untreated seeds 16 h after treatment. *P* values were calculated on the basis of the Fisher method using nominal *P* values provided by edgeR and DEseq2. Dashed lines represent the threshold defined for *P* (horizontal) and fold change (vertical) for a gene to be considered as DEG. Tags label the genes related to seed germination progress: OEE1, oxygen-evolving enhancer protein 1; OEE3-2, oxygen-evolving enhancer protein 3-2, chloroplastic; OEE2-1, oxygen-evolving enhancer 2-1, chloroplastic; Fd-like, erredoxin-like; PSAD2, photosystem I reaction centre subunit II, chloroplastic; Gdh, glutamate dehydrogenase; rbcL, ribulose bisphosphate carboxylase small chain; ppdK, pyruvate, phosphate dikinase; pckG, phosphoenolpyruvate carboxykinase; Clo, caleosin; Hsp70_2, heat shock 70 kDa protein_2; Hsp, class I heat shock protein; Hsp70, heat shock 70 kDa protein_1; and Hsp22, 22.0 kDa class IV heat shock protein. NS, not significant. **c**, Molecular families corresponding to fatty acyls analogous to triacylglycerides and glycerophospholipids differentially abundant in seeds 0 h after seed bacterization. Pie charts inside the nodes indicate the mean of the peak abundance of each metabolite in the corresponding condition. Node shape indicates the level of identification according to ref. [78]. The chemical structures of annotated features based on spectral matches to GNPS libraries are also presented for each molecular family. POPC, 1-palmitoyl-2-oleoyl-glycero-3-phosphocholine; PC, phosphocholine. **d**, Adult plants grown from seeds treated with *B. subtilis* (3610, right) or from untreated seeds (control, left). **e**, Circos plot showing the top 100 metabolites significantly more abundant in leaves of plants grown from bacterized seeds versus those in leaves of the control plants. Ribbon colours refer to the class of each metabolite, and thickness is proportional to the log$_2$FC values. The curved colour bar shows the contribution of each metabolite (rectangles inside the bar coloured according to their chemical class) to the total abundance (100%), ordered from highest to lowest contribution.

compound analogues of sphinganines and an increase in abundance of the prenol lipid and stilbene molecular families (Extended Data Fig. 4b). The presence of a functional ECM in the WT strain triggered the accumulation of fatty acyls belonging to two different clusters, and treatments with ECM mutants produced a clear decrease in organooxygen compounds and accumulation of specific prenol lipids (Extended Data Fig. 4b).

**Lysophospholipids, glutathione and growth promotion.** According to our data, TasA and fengycin are highly relevant molecules of the ECM with regards to growth promotion of *B. subtilis* in seeds. TasA, in common with other amyloids, is a polymorphic protein that can adopt a variety of structural conformations that have different biochemistries and functions[6,22,23]. We polymerized purified homogeneous TasA monomers to show that large aggregates

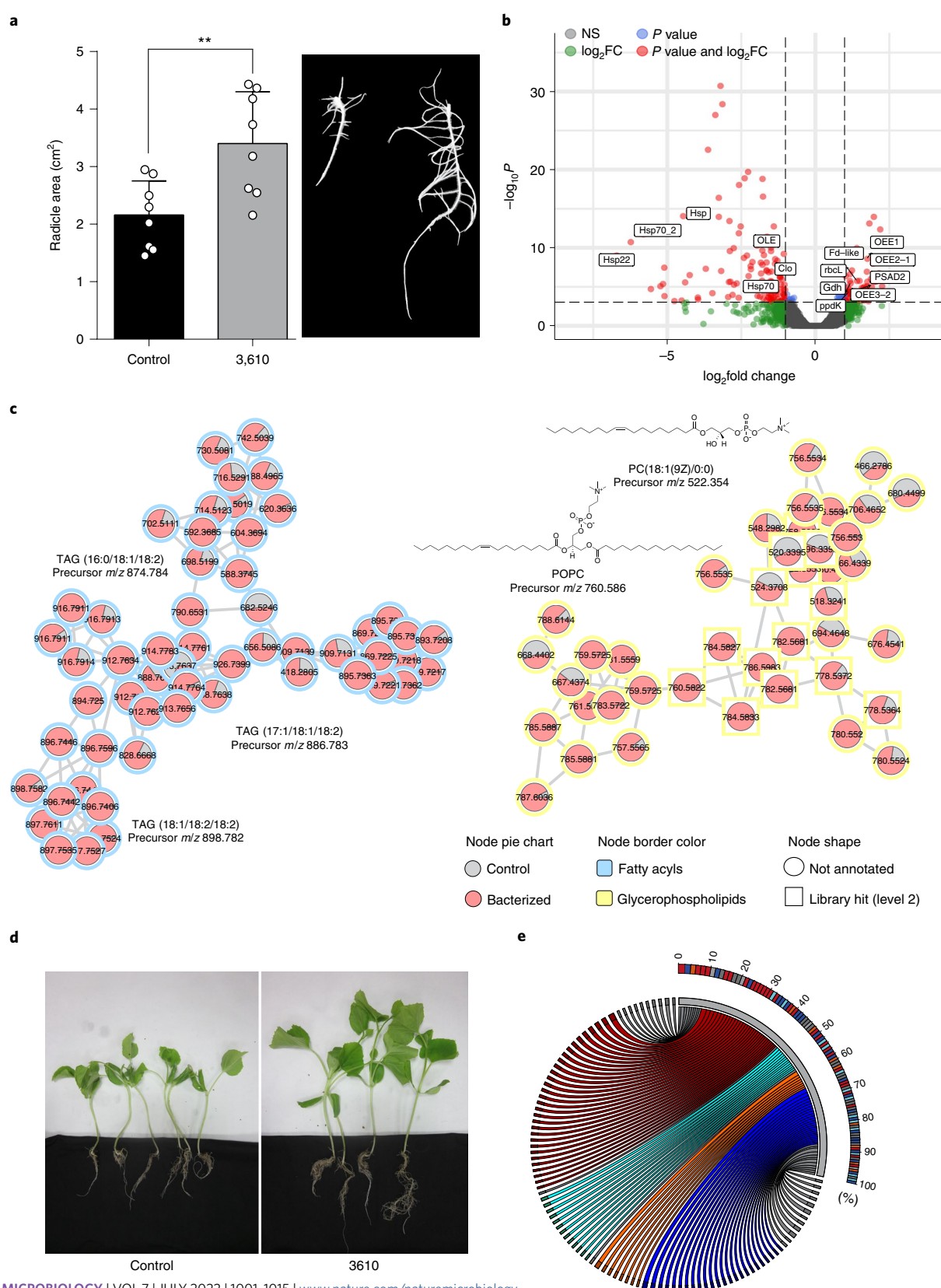

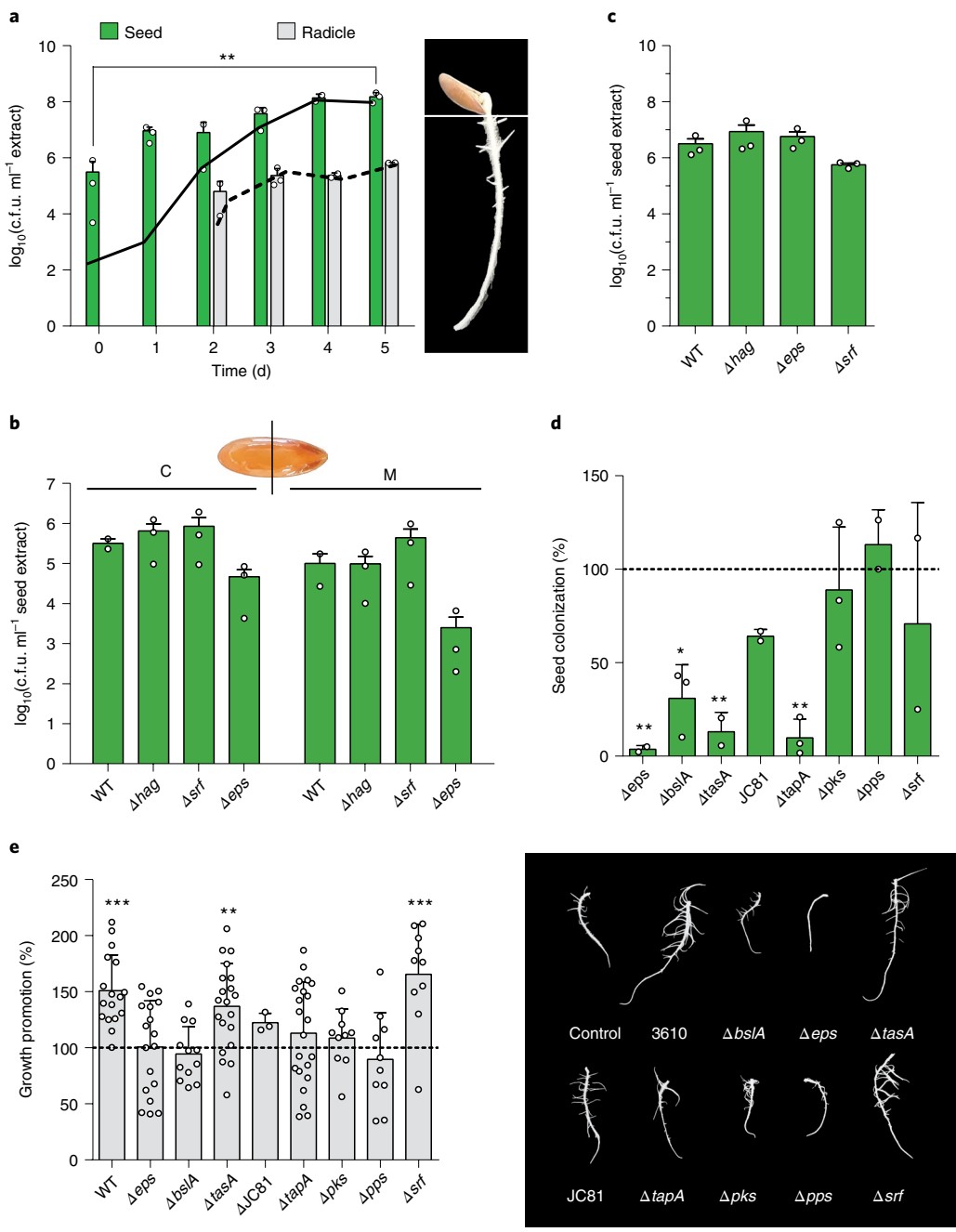

**Fig. 2 | The extracellular matrix of *B. subtilis* is required for efficient seed colonization and radicle growth stimulation. a**, *B. subtilis* dynamics (c.f.u. counts) in seed and radicle extracts during the first 5 d after seed treatment. Bars represent average values ± s.e.m. of total c.f.u. in seeds and radicles, and solid and dashed lines represent c.f.u. corresponding to the number of spores in seeds and radicles, respectively. Statistical significance was assessed by a one-tailed *t*-test between total c.f.u. at initial and final timepoints ($n=3$, \*\*$P=0.0062$). **b**, C.f.u. counts of extracts from sectioned bacterized seeds separated at the micropylar (M) and chalazal (C) regions 4 h after treatment with the WT and motility-related mutants. Average values ± s.d. are shown ($n=3$). **c**, C.f.u. counts of total extracts from bacterized seeds 4 h after treatment with the WT and motility-related mutants. Average values ± s.d. are shown ($n=3$). **d**, Seed colonization of the ECM mutants relative to that of the WT bacteria (assumed to be 100%) measured in seed extracts 5 d after seed treatment. Average values ± s.e.m. are shown. Statistical significance was assessed by one-way ANOVA with Dunnett's multiple comparisons test (each treatment vs WT treatment) ($n=3$, \*\*$P = 0.0027$ in Δeps, \*$P = 0.0135$ in ΔbslA, \*\*$P = 0.0067$ in ΔtasA, \*\*$P = 0.0013$ in ΔtapA). **e**, Left: percentage of radicle growth increase in seeds treated with WT cells or ECM mutant cells (5 d after seed treatment) normalized to the average radicle area of untreated seeds (100%, dashed line). Mean ± s.d. Statistical significance was assessed by one-way ANOVA with post hoc Dunnett's multiple comparisons test (each treatment vs control treatment) ($n=17$ in control and 3610, $n=18$ in Δeps, $n=12$ in ΔbslA, $n=21$ in ΔtasA, $n=3$ in JC81, $n=22$ in ΔtapA, $n=10$ in Δpks and Δpps, $n=13$ in Δsrf; \*\*\*$P = 0.0002$ in WT, \*\*$P = 0.0076$ in ΔtasA, \*\*\*$P < 0.0001$ in Δsrf). Right: representative radicles from untreated seeds and seeds treated with WT or ECM mutant cells 5 d after treatment.

stimulate radicle growth (Extended Data Fig. 5a). Moreover, fresh apoplast fluid extracted from melon seeds promoted the polymerization of TasA into aggregates (Extended Data Fig. 5b), indicating

that this largely active polymerized form of TasA may predominate inside seeds. Evaluation of stimulatory activity showed that a 3 μM solution of the most active form of TasA or a 10 μM solution of puri-

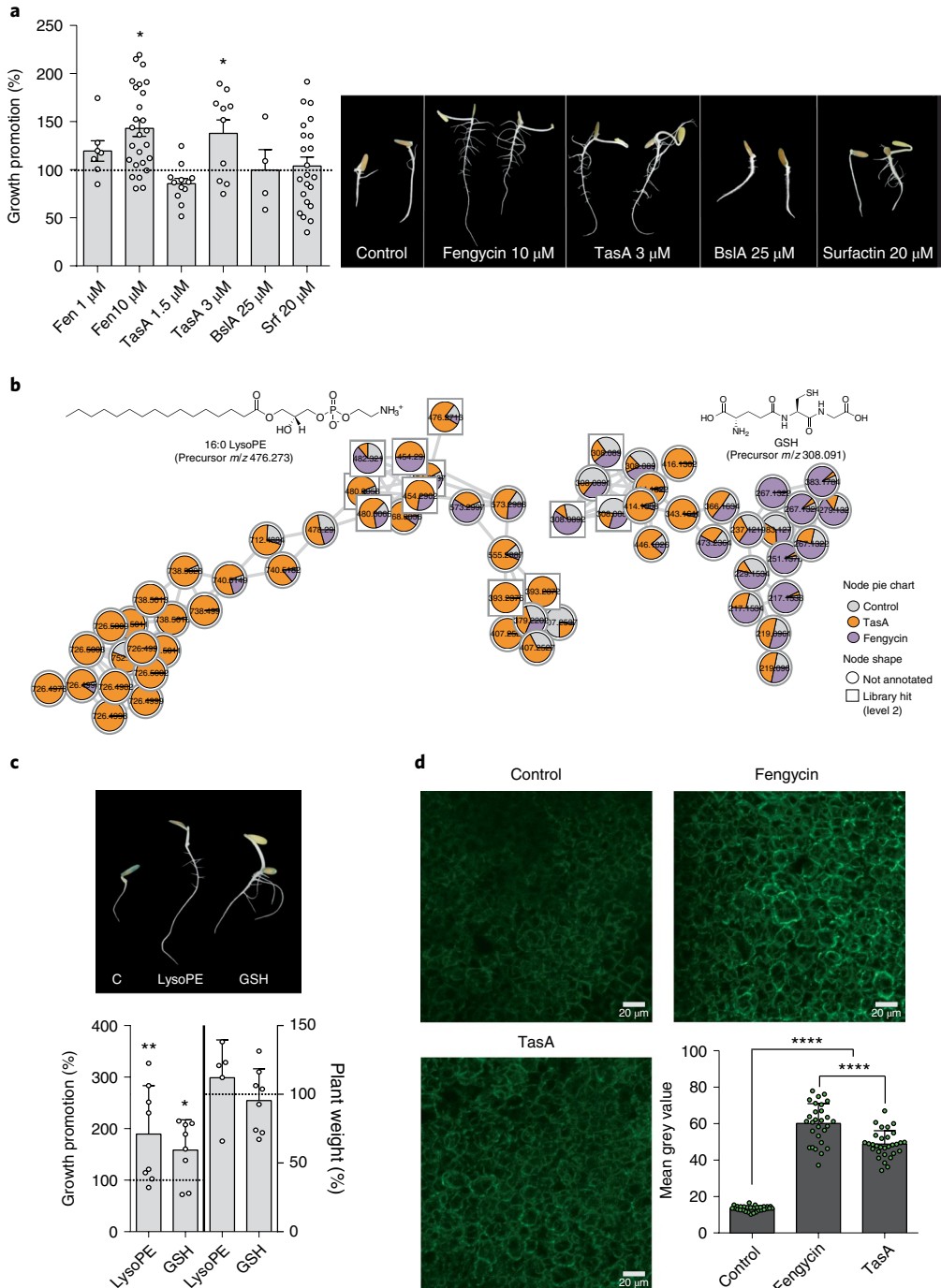

**Fig. 3 | Amyloid TasA and fengycin stimulate radicle development. a**, Left: percentage of radicle growth promotion of seeds treated with purified ECM components normalized to the average radicle area of untreated seeds (100%, dotted line) 5 d after treatments. Average values ± s.d. are shown. Statistical significance was assessed by one-way ANOVA with post hoc Dunnett's multiple comparisons test (each treatment vs control) ($n = 64$ in control, $n = 8$ and $n = 27$ in Fengycin 1 μM and 10 μM, respectively, $n = 12$ and $n = 10$ in TasA 1.5 μM and 3 μM, respectively, $n = 4$ in BslA 25 μM and $n = 22$ in surfactin 20 μM; *$P = 0.0104$ in Fen10 μM, *$P = 0.0313$ in TasA 3 μM). Right: representative radicles of untreated seeds or seeds treated with purified ECM 5 d after treatments. **b**, Molecular families corresponding to LysoPE and GSH. Pie charts represent the mean peak abundance of metabolites in each condition. Node shape indicates the level of identification according to ref. [78]. **c**, Top: representative radicles treated with water (C), LysoPE and GSH 5 d after treatment. Bottom: left Y axis, percentage of radicle growth promotion of seeds treated with water or purified LysoPE and GSH normalized to the control radicle area (100%, dotted line) 5 d after treatment. Statistical significance was assessed by one-way ANOVA with post hoc Dunnett's multiple comparisons test (each treatment vs control) ($n = 24$ in control, $n = 8$ in LysoPE and GSH; *$P = 0.0451$ in GSH, **$P = 0.0021$ in LysoPE); right Y axis, percentage of weight of adult plants grown from seeds treated with LysoPE and GSH, normalized to the weight of control plants (100%, dotted line) ($n = 12$ in control, $n = 5$ in LysoPE and $n = 8$ in GSH). Average values ± s.d. are shown. **d**, Representative CLSM images of the inner tissues of dihydrorhodamine-stained seeds immediately after seed treatments with water (control), fengycin or TasA. Graph represents the intensity of the fluorescence in each condition, measured as the mean grey value from $n = 30$ random sections from 3 different fields. Average values ± s.d. are shown. Statistical significance was assessed by one-way ANOVA with multiple comparisons test ($P < 0.0001$).

fied fengycin significantly increased the area of the radicles compared with untreated seeds (Fig. 3a). The lack of stimulatory activity of purified BslA or surfactin confirmed our previous findings using *B. subtilis* mutants (Figs. 2e and 3a).

The metabolome of individual radicles ($n=3$) that emerged from seeds treated with fengycin or TasA was analysed to define their contributions to the metabolic signatures associated with the promotion of radicle growth. Results of principal component analysis and heatmap analyses indicated sample clustering in three groups (Extended Data Fig. 6a,b). Statistical analysis by partial least-squares discriminant analysis indicated that glycerophospholipids, fatty acyls, organooxygen compounds or carboxylic acids were discriminating metabolites in the case of TasA or fengycin treatments, and feature-based molecular networking of these features showed integration into molecular families composed of other metabolites that generally followed the same abundance patterns (Extended Data Fig. 6c–e). Further refinement of this analysis identified two molecular families of interest: lysophosphatidylethanolamine (LysoPE), mainly associated with TasA treatment, and an analogue of reduced glutathione (GSH) that accumulated in the radicles grown from seeds treated with fengycin (Fig. 3b). The treatment of seeds with commercially available LysoPE or GSH increased radicle growth as much as treatment with fengycin or TasA, but did not produce the long-term promoting effects observed in plants 1 month after treatment of seeds with B. *subtilis* (Fig. 3c).

GSH is an essential and polyvalent metabolite with special antioxidant functions, which maintains cellular redox homoeostasis and development, growth or response of plants to a variety of stimuli[24]. Further, an increase in the total pool of glutathione usually occurs 3 to 4 d after ROS exposure[24,25], therefore we hypothesized that one possible outcome of fengycin treatment might be ROS production. The level of ROS in seeds after treatment with fengycin was statistically higher than that in untreated seeds. An intermediate increase in ROS, which was statistically less than that elicited by fengycin, was also observed in seeds treated with TasA (Fig. 3d).

**TasA and fengycin differentially target seed oil bodies.**
Accumulation of bacterial cells in the micropylar endosperm (Extended Data Fig. 3b), accumulation of TAGs (Fig. 1c) and changes in the lipid composition of the radicles and adult plants after treatment of seeds with bacteria or ECM components (Extended Data Figs. 2b and 6d,e) led us to hypothesize that seed oil bodies (OBs), which are nutrient reservoir organelles mainly composed of TAGs and other neutral lipids, are the target of fengycin and TasA. During germination, OBs are degraded by interaction with glyoxysomes that feed the plant embryo[26,27]. We applied CLSM to thin sections of seeds treated with a suspension of *B.* subtilis or untreated seeds using a neutral lipid stain 16 h after seed treatment and found an abundance of OBs in the endosperm of untreated seeds. In agreement with our hypothesis, certain regions of the endosperm

accumulated high levels of disaggregated oil bodies that were surrounded by bacterial aggregates (Extended Data Fig. 7a).

Treatment with purified TasA or fengycin promoted in vivo disaggregation of OBs, similar to the effect observed after treatment with *Bacillus* cells (Extended Data Fig. 8a). Transmission electron microscopy (TEM) analysis of thin sections of negatively stained seeds showed OB disaggregation and localization of OB-surrounding glyoxysomes in samples treated with TasA (Extended Data Fig. 7b,c). In fengycin-treated seeds, the most predominant changes were disaggregation of OBs and the presence of empty areas between the cell wall and the cytoplasm, suggestive of plasmolysis, a cellular response previously observed in the interaction of fengycin with membranes of fungal cells[28] (Extended Data Fig. 7c). The CLSM analysis of OBs purified from melon seeds treated with TasA or fengycin and double-stained with Fast Green FCF and Nile red indicated differences in the size and pattern of aggregation of OBs: fengycin induced disaggregation of OBs as observed–in vivo, while TasA preferentially induced a higher level of aggregation of individual OBs than that observed in vivo (Fig. 4a). Higher intensity of the green fluorescence signal near OB aggregates in the samples treated with TasA suggested non-specific staining of the amyloid protein and its localization around the vesicles. TEM analysis of purified OBs and immunogold labelling with anti-TasA antibodies indicated the presence of gold particles decorating the perimeter of the OBs and interconnecting TasA fibres (Extended Data Fig. 7d). The results of pulldown assays of whole protein extracts of the seeds indicated co-elution of purified TasA with oleosins, which are structural proteins that modulate the stability and size of OBs, regulate lipid metabolism and have an important role in OB degradation[29,30] (Fig. 4b and Supplementary Table 5). Interaction of the two proteins was confirmed by far-western blot analysis of purified TasA and purified oleosin 1 from *Arabidopsis thaliana* (Fig. 4c).

We propose that TasA–oleosin interaction is responsible for the aggregation of OBs, favouring the accumulation of lysophospholipids after the optimization of lipid catabolism in glyoxysomes. Fengycin has been described to efficiently interact with and disrupt artificial lipid monolayers or bilayer membranes in a concentration-dependent manner[31]. In addition to micellar concentrations of fengycin, specific disruption of the membranes relies on the lipid composition of the target[32]. Exact chemical composition of phospholipids of the membranes of OBs in the seeds is not known and varies between species[33]; however, we propose that high affinity of fengycin for lipid membranes explains disaggregation and a reduction in the size of OBs.

Non-targeted metabolomic analyses provided additional information about the change in relative abundance of metabolites and potential (bio)chemical modifications of seed metabolites from 0 to 24 h after treatment with TasA or fengycin. The use of the chemical proportionality tool (ChemProp[34]) showed putative changes in the dynamics of TAGs analogues after seed treatments (Fig. 4d). In

---

**Fig. 4 | Differential interactions of amyloid TasA and fengycin with OBs are related to their specific stimulatory activity on plant growth. a**, CLSM images of purified OB suspension 16 h after the addition of 10 μM fengycin or 3 μM TasA. Scale bars, 5 μm. OBs were stained with Nile red and membrane proteins were stained with Fast Green FCF. Colour bars below each image indicate the percentage of various statuses of OBs calculated by measuring each particle area (black, aggregation (area > 5 μm²); red, individual OBs (0.5 μm² < area < 5 μm²); grey, disaggregated OBs (area < 0.5 μm²)) in representative CLSM fields of view. **b**, Coomassie brilliant blue (CBB)-stained SDS–PAGE gel of elution fractions (E3 and E4) of pulldown assay of TasA with proteins in the extracts from melon seeds (left) and in absence of TasA as bait protein (right). The co-eluting band is marked with red arrows and identified as an oleosin by HPLC-ESI-MS/MS analysis. TasA bands are marked with red asterisks. **c**, Far-western blotting (F WB). Left: a CBB-stained SDS–PAGE gel of fractions of oleosin obtained during purification; BSA as a control, and fractions of BslA and TasA that were obtained during purification. Middle: immunoblot of purified proteins using an anti-TasA antibody (1:20,000). Right: far-western blot using an anti-TasA antibody (1:20,000) after renaturing of the proteins and incubation with TasA protein as a bait. **d**, Chemical proportionality analysis of the TAGs analogues molecular family in non-treated seeds (control) and seeds treated with fengycin or TasA from 0 h to 24 h after treatment. The size of the nodes is directly related to the abundance of the features at time 0 h. Arrows indicate chemical directionality of the modifications found and the colour scale of the arrow represents the ChemProp score. Floating bar plots represent the peak abundances of three selected TAGs analogues in non-treated (C) and treated seeds with fengycin (F) or TasA (T) at 0 h and 24 h after treatment. Average values are shown, with the upper and lower borders of each bar representing maximum and minimum values, respectively ($n=3$). In

control seeds, many modifications in this molecular family denoted an important catalytic activity occurring over key storage metabolites in this time period. Treatments with TasA and, more notably, with fengycin led to an increase in the initial abundance of many metabolites (Fig. 4d, floating bar plots) that would reflect the faster metabolic activity of these seeds. A decrease in the Chemprop score

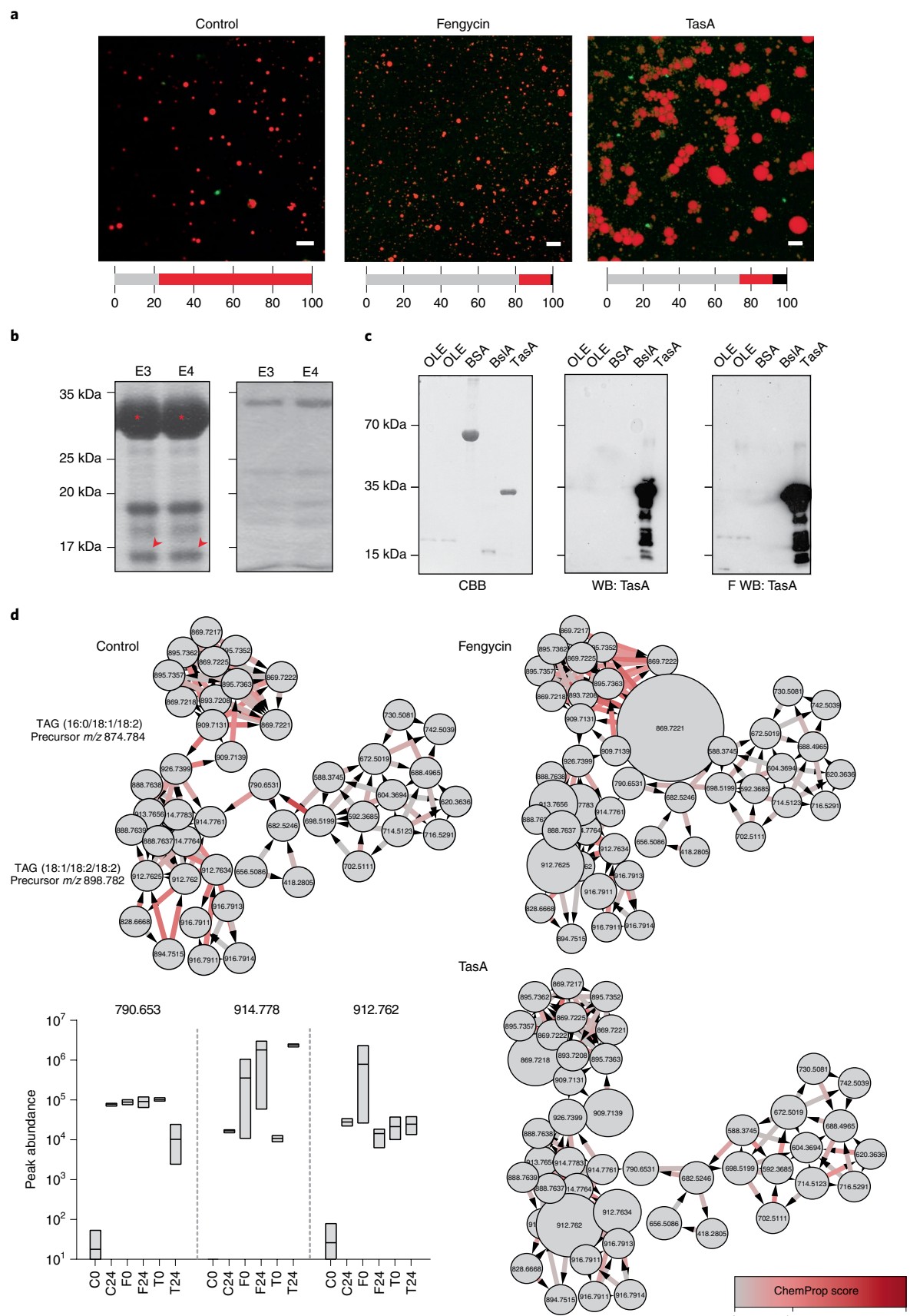

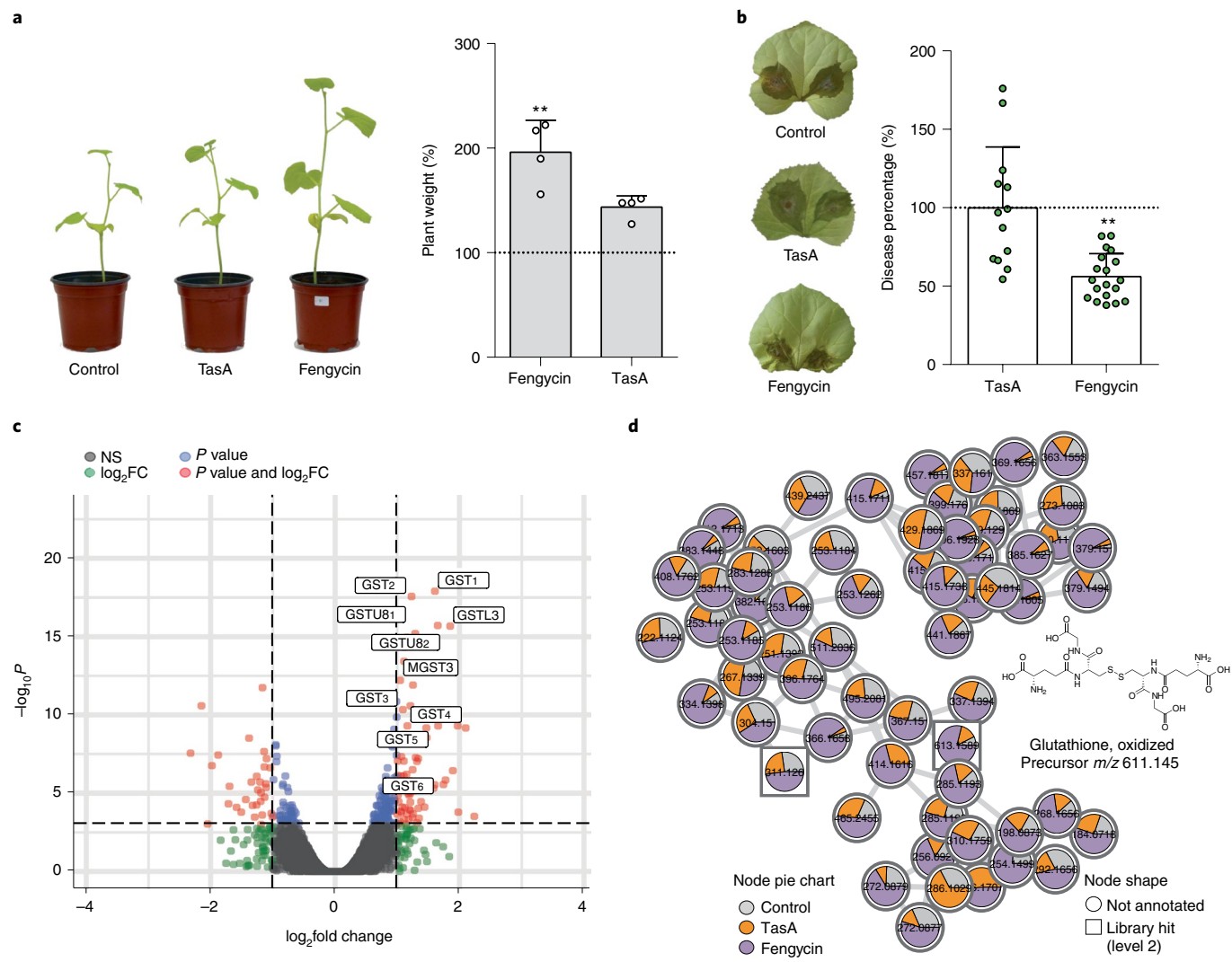

**Fig. 5 | Seed treatment with fengycin stimulates the growth and immunization of adult plants. a**, Left: representative adult plants grown from seeds treated with water (control), fengycin or TasA. Right: percentage of the weight of plants grown from seeds treated with water (control), 10 μM fengycin or 3 μM TasA normalized to the average weight of plants grown from control seeds (100%, dotted line). Average values ± s.d. are shown. Statistical significance was assessed by one-way ANOVA with post hoc Dunnett's multiple comparisons test (each treatment vs control treatment, $n = 4$, **$P = 0.0021$). **b**, Left: necrotic symptoms in leaves of adult plants grown from seeds treated with water (control), fengycin or TasA 72 h after treatment with *B. cinerea* spores. Right: percentage of disease calculated by measuring the lesion areas and normalizing to the average lesion area of the control leaves (100%, dotted line). Average values ± s.d. are shown. Statistical significance was assessed by one-way ANOVA with post hoc Dunnett's multiple comparisons test (each treatment vs control treatment) ($n = 28$ in control, $n = 13$ in TasA and $n = 19$ in Fengycin; **$P < 0.0043$). **c**, Volcano plot representation of DEGs identified by total transcriptome analysis in leaves of plants grown from control seeds or seeds treated with fengycin 48 h after treatment with *B. cinerea* spores. $P$ values were calculated on the basis of the Fisher method using nominal $P$ values provided by edgeR and DEseq2. Tags label the genes related to glutathione metabolism: GST$_{1,2,3,4,5,6}$ (glutathione S-transferases 1 to 6), GSTL3 (glutathione S-transferase L3-like), GSTU8$_{1,2}$ (glutathione S-transferases U8-like 1 and 2) and MGST3 (microsomal glutathione S-transferase 3). Dashed lines represent the threshold defined for $P$ (horizontal) and fold change (vertical) for a gene to be considered as DEG. **d**, Molecular family of oxidized glutathione and related metabolites differentially abundant in aerial regions of seedlings 120 h after seed treatment with fengycin. Pie charts indicate the mean peak abundance of metabolites in each condition. Node shape indicates the level of identification according to ref. [78].

indicated that the significance of the changes between 0 h and 24 h would be directly related with the earlier initiation of the catalytic activity over these molecules. Similar changes to TAGs were also remarkable in the dynamics of glycerophospholipids (Extended Data Fig. 8b).

**Fengycin promotes plant growth and resistance to *Botrytis cinerea*.** Plants that emerged from seeds treated with fengycin were larger than plants grown from untreated or TasA-treated seeds (Fig. 5a). Adult plants grown from seeds treated with LysoPE or GSH,

however, did not show a significant increase in growth over time compared with plants from untreated seeds (Fig. 3c). These findings indicate that the presence of different signalling events underpin short-term radicle growth mediated by TasA, fengycin or associated LysoPE and GSH molecules, as well as long-term growth of adult plants specifically associated with fengycin. GSH associated with treatments of the seed with fengycin did not produce a sustained long-term growth-promoting effect, suggesting that a constant endogenous trigger is required that can only be achieved by fengycin treatment. Both TasA and fengycin increased the level of

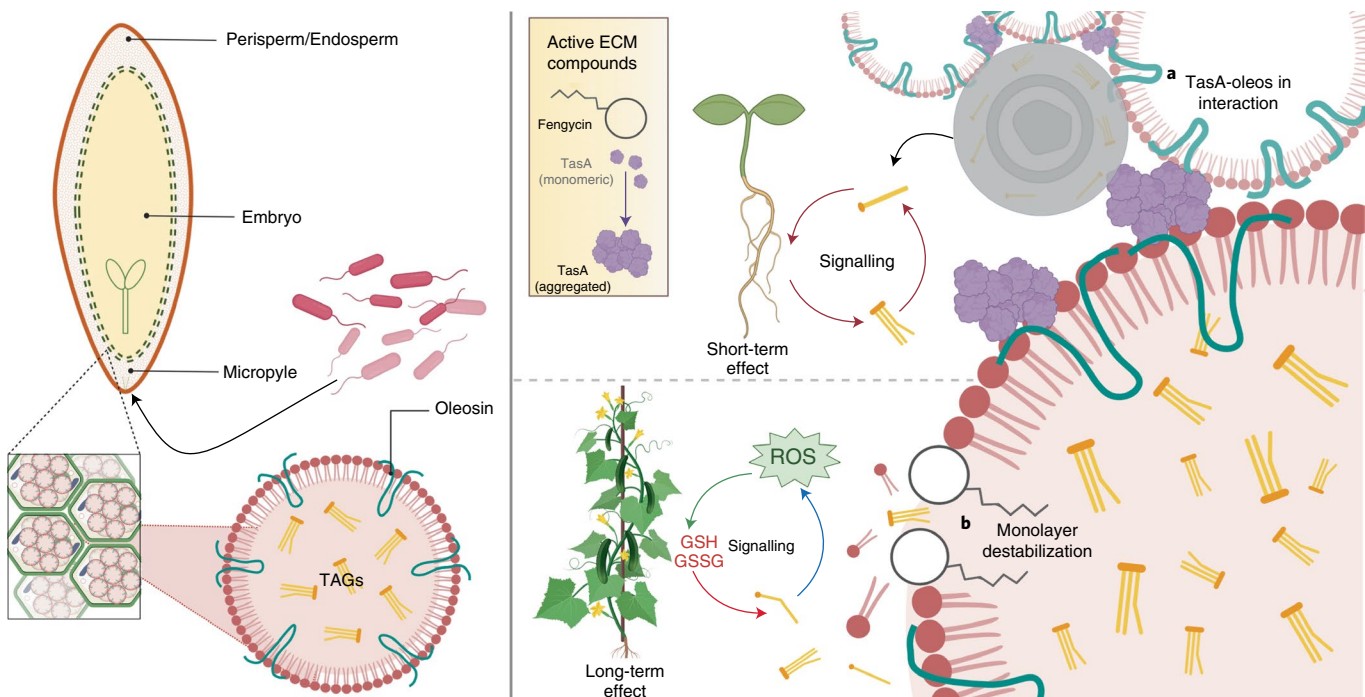

**Fig. 6 | Overall scheme of the proposed mechanism of growth-promoting activities of TasA and fengycin over seeds.** Left: structure of melon seeds and OBs. The perisperm/endosperm of the seeds is enriched in storage cells containing abundant OBs, which are composed of a monolayer of phospholipids and structural proteins, mainly oleosins, and contain mostly triacylglycerides. After the treatment, *B. subtilis* cells passively enter the inner tissues of the seeds through the micropyle. Right: **a**, Inside the seeds, aggregates of TasA interact with oleosins of OBs, favouring the aggregation of these vesicles around glyoxysomes, interacting with organelles where their content is degraded to release stored energy. This process would lead to the accumulation of lysophospholipids involved in signalling related to a short-term effect (radicle growth). **b**, Fengycin induces the accumulation of stimulating levels of ROS and destabilizes the OB membrane, triggering the accumulation of specific lipid molecules and GSH, an antioxidant molecule implicated in long-term stimulation of adult plant growth and immunization. GSSG, oxidized glutathione. Created with BioRender.com.

ROS inside seeds, but the highest response was elicited by fengycin (Fig. 3d). ROS production is a normal response after the first water uptake during seed imbibition and, in a certain range of concentrations, ROS has a well-recognized role in endosperm weakening, protection against pathogens, mobilization of seed reserves and interaction with plant hormones during the process of seed germination[35–37]. These results suggest that the increase in ROS levels after seed treatment with fengycin, but not TasA, may act as a beneficial stimulus for further plant development mediated by GSH accumulation.

Previous studies revealed that application of fengycin elicits plant defence responses in adult plants[7,38]. Therefore, we investigated whether treatment of seeds with fengycin immunized adult plants against aboveground pathogens by inoculating the third leaf of adult plants with a spore suspension of *Botrytis cinerea*. The size of necrotic lesions induced by the fungus in plants grown from fengycin-treated seeds was significantly smaller than that of control plants or plants grown from TasA-treated seeds (Fig. 5b). Transcriptomic analyses of infected leaves revealed a high basal level of transcripts, some of them related to plant defence mechanisms such as Allene oxide cyclase/synthase, directly implied in the biosynthesis of jasmonic acid[39] (Supplementary Table 5) in leaves of plants emerged from seeds treated with fengycin. Their expression was maintained or increased 48 h after inoculation with *B. cinerea*, while an increase was found in adult plants from non-treated seeds as a result of the response to the infection. Second, upregulation of the genes related to glutathione metabolism, specifically including glutathione S-transferases (GSTs), occurred 48 h after the challenge with *B. cinerea* (Fig. 5c and Extended Data Fig. 9a). Participation of GSTs in antioxidant reactions, together with the important cellular antioxidant GSH, has been proven to mitigate oxidative stress caused by necrotrophic fungus *B. cinerea* in infected tissues[40]. Metabolomic data from the radicles of fengycin-treated seeds demonstrated an increase in the levels of certain molecules of the same cluster that included GSH (Fig. 3b), and adult plants grown from bacterized seeds accumulated certain molecules belonging to the GSH cluster, especially in the aerial region (Extended Data Fig. 9b). In addition, metabolomes of aerial parts of seedlings 5 d after treatment with fengycin also showed an increase in oxidized glutathione and glutathione-related metabolites in addition to defence-related compounds such as flavonoids (that is, mangiferin and 2-O-rhamnosilvitexin) (Fig. 5d and Extended Data Fig. 9c).

## Discussion

In this study, we demonstrated that in addition to a structural role of bacterial ECM in biofilm formation, it is also essential for *Bacillus* colonization of melon seeds and growth promotion, including induction of anti-fungal resistance in the adult plant. Our findings indicate that the beneficial effects of *B. subtilis* treatment on melon seeds are associated with changes in the initial ROS increase during seed imbibition, and the contents and specific pools of metabolites released from storage tissues, which were mediated by at least two ECM components, fengycin and TasA.

We propose that specific interactions of TasA or fengycin with OBs defines two different physiological responses. First, a short-term effect on radicle growth, mediated by both components and related to changes in the dynamics of TAGs after differential targeting of oil bodies and lipid catabolism optimization. On one hand, the oleosin–TasA interaction determines the OB aggregations around glyoxysomes and the subsequent accumulation of

lysophospholipids acting as signal molecules for growth. On the other hand, the OB disaggregations caused by the action of fengycin over membranes determine a faster mobilization of their contents. Second, a long-term effect of growth promotion that persists in adult plants and is mediated by fengycin, through an initial increase in ROS that causes accumulation of GSH that is maintained over time and makes the plant more tolerant to biotic stresses (Fig. 6).

Glycerophospholipids have both structural and signalling roles in plants, and this bifunctionality is in part due to continuous synthesis and turnover of endogenous pools[41]. Accumulating evidence suggests a role for lysophospholipid derivatives in signalling processes in plant cells. LysoPE treatment has been reported to delay fruit softening when used postharvest, mitigate defoliation effects of ethephon, and delay leaf and fruit senescence in tomato and potato[42,43] and is used commercially as a plant bioregulator to improve plant product quality[41,44]. On the other hand, glutathione is an essential metabolite with antioxidant properties, and is involved in cellular redox homoeostasis and many other physiological processes such as the development, growth and environmental response of plants[24]. The accumulation of glutathione-related metabolites in radicles and aerial parts of seedlings, and the higher expression of GSTs in leaves after seed treatment with fengycin, suggest that initial treatment causing an increase in ROS might act as a beneficial stimulus for plant development, and confer enhanced antioxidant capacity to mitigate an imbalance in the redox status in adult plants imposed by infection with *B. cinerea*.

We propose that only those seeds containing abundant OBs, and characterized by specific morphology in their primordial tissues, respond to the beneficial interaction with *B. subtilis*, and that this interaction is mediated at least in part by fengycin and TasA. According to this model, wheat or maize, which are monocotyledonous plants whose seeds are composed of starchy endosperm surrounded by a layer of living cells, did not react to the presence of fengycin, whereas cucumber or soybean seeds, which are anatomically different but do contain OBs, responded with a growth-promoting effect (Supplementary Table 6).

We note however that *B. subtilis* communities (cells and ECM) still retain stimulatory activity in these and other seeds plants, so it cannot be excluded that additional plant pathways or *Bacillus*-derived structural or chemical components, such as secondary metabolites or phytohormones, are specifically involved in maintaining these intricate interkingdom mutualistic interactions.

## Methods

**Strains, media and culture conditions.** The bacterial strains used in the present study are listed in Supplementary Table 1. Bacterial cultures were grown at 37 °C from frozen stocks on lysogeny broth (LB: 1% tryptone (Oxoid), 0.5% yeast extract (Oxoid) and 0.5% NaCl) plates. Isolated bacteria were inoculated in appropriate media. The necrotrophic fungus *B. cinerea* was grown at 25 °C from a frozen stock in potato dextrose agar (PDA) plates and maintained until inoculum preparation. *Escherichia coli* DH5α was used for cloning and plasmid replication. *E. coli* BL21(DE3) and BL21(AI) were used for protein purification. The final antibiotic concentrations for *B. subtilis* were: MLS (1 µg ml⁻¹ erythromycin and 25 µg ml⁻¹ lincomycin), spectinomycin (100 µg ml⁻¹), tetracycline (10 µg ml⁻¹), chloramphenicol (5 µg ml⁻¹) and kanamycin (10 µg ml⁻¹).

**Quantitative analysis of bacterial growth and promotion of radicle growth.** Bacterial strains were grown in liquid LB at 30 °C overnight. The cells in the cultures were washed twice with sterile distilled water, and bacterial cell suspensions were adjusted to the same optical density (OD₆₀₀). When using a fluorescent inoculum, CellTracker CM-DiI dye (Invitrogen, C7000) was added to the bacterial suspension. Melon seeds (Rochet Panal - Fitó) were surface-sterilized with 0.1% sodium hypochlorite and bacterized with bacterial cell suspensions by bathing for 1 h at room temperature. The seeds were placed in Petri dishes with wet filter paper, covered with foil and maintained in growth chambers at 25 °C for 5 d. For quantitative analysis of the bacteria, the radicles were separated from the seeds. The radicles were placed in 1.5 ml microcentrifuge tubes with 1 ml PBS medium and 1 g of 3-mm-diameter glass beads, and attached bacteria were recovered by vortexing for 1 min. The seeds were ground using a mortar and a pestle in 1 ml PBS. Total c.f.u. were assayed by plating serial dilutions of resulting suspensions on

solid LB medium. C.f.u. of the spores were assayed using the same suspension after thermal shock at 80 °C for 5 min. Seeds treated with water were used as controls.

Analysis of radicle growth-promoting effects was based on radicle areas measured 5 d after seed treatment using ImageJ software. The medium radicle area of the control treatment group was assumed as 100% growth and was used to normalize the percentage of growth-promoting effect corresponding to strain treatment groups. Treatments with compounds purified from the extracellular matrix were performed in a similar manner. BslA and TasA proteins were purified as described above while fengycin, surfactin and iturin were commercially acquired (Biosynth-Carbosynth AF35268, Sigma-Aldrich S3523 and I1774, respectively). Unless indicated, TasA was used at 3 µM and fengycin at 10 µM. Iturin was used for seed treatments at 260 µM, and lysophosphoethanolamine 16:0 (Avanti Polar Lipids, 856705) and reduced glutathione (Sigma-Aldrich G4251) were used at 0.2 mg ml⁻¹ and 0.25 mM, respectively. For the evaluation of long-term growth promotion, seedlings from treated seeds were transferred to pots and maintained in a growth chamber at 25 °C.

**Protein cloning and purification.** TasA protein was purified as previously described[6]. BslA protein was purified from freshly transformed *E. coli* BL21-AI cells expressing the pDEST17 plasmid harbouring the *bslA* gene, which was initially subcloned into the donor vector PDONR207 using a Gateway recombination system. The primers designed to amplify the *bslA* gene were as follows: bslAFw: 5'GGGGACAAGTTTGTACAAAAAAGCAGGCTCAGCTG-GAATCTACATCAACTAAAGC3'; bslARv: 5'GGGGACCACTTTGTACAAGA AAGCTGGGTTTAGTTGCAACCGCAAGGCT3'. The colonies of *E. coli* BL21-AI were picked and resuspended in 500 ml LB containing 100 µg ml⁻¹ ampicillin and incubated at 37 °C in an orbital shaker to reach an OD₆₀₀ of 0.7. Then, the expression of recombinant protein was induced by 0.2% L-arabinose. After 4 h, the cells were collected and resuspended in buffer A (20 mM sodium phosphate, 500 mM NaCl and 20 mM imidazole) containing 400 µl 0.1 M phenylmethylsulfonyl fluoride, 40 µl lysozyme and 4 µl Cellytic 10x (Sigma-Aldrich). After a 45 min incubation with shaking at room temperature, the cells were disrupted by sonication (3 pulses of 1 min at an amplitude of 80%). The cell lysate was centrifuged at 12,000 g for 15 min at 20 °C, and the supernatant was filtered through a 0.45-µm-pore filter. The protein was purified using an AKTA Start fast protein liquid chromatography system (GE Healthcare). The soluble fraction was loaded into a 5 ml HisTrap HP column (GE Healthcare) previously equilibrated with buffer A. The protein was eluted from the column with elution buffer (20 mM sodium phosphate, 500 mM NaCl and 500 mM imidazole, pH 8). After affinity chromatography, the purified protein was loaded into a HiPrep 26/10 desalting column (GE Healthcare), and the buffer was exchanged for 20 mM Tris and 50 mM NaCl.

Oleosin protein was purified from inclusion bodies using freshly transformed *E. coli* Lemo21(DE3) cells expressing the pDEST17 plasmid harbouring the sequence of OLEOSIN1 from *Arabidopsis thaliana*, which was initially subcloned in the donor vector pENTR223 acquired from Arabidopsis Biological Resource Center using a Gateway recombination system. *E. coli* Lemo21(DE3) colonies were picked and resuspended in 500 ml LB containing 100 µg ml⁻¹ ampicillin, 5 µg ml⁻¹ chloramphenicol and 100 µM rhamnose, and incubated at 37 °C with orbital shaking to reach an OD₆₀₀ of 0.4. Then, the expression of the recombinant protein was induced by 400 mM isopropyl β-D-1-thiogalactopyranoside (IPTG). After 3 h, the cells were collected by centrifugation (5,000 g, 15 min, 4 °C), resuspended in buffer A (50 mM Tris and 150 mM NaCl, pH 8) and then centrifuged again. The pellets were stored at −80 °C until purification or processed after 15 min. After thawing, the cells were resuspended in buffer A, sonicated on ice (3 × 45 s, 60% amplitude) and centrifuged (15,000 g, 60 min, 4 °C). The supernatant was discarded because the protein was mainly expressed in inclusion bodies. The pellet was resuspended in buffer A supplemented with 2% Triton X-100, incubated at 37 °C with shaking for 20 min and centrifuged (15,000 g, 10 min, 4 °C). The pellet was extensively washed with buffer A, centrifuged (15,000 g, 10 min, 4 °C), resuspended in denaturing buffer (50 mM Tris, 500 mM NaCl and 6 M GuHCl) and incubated at 60 °C overnight for complete solubilization. The lysates were clarified by sonication on ice (3 × 45 s, 60% amplitude) and centrifugation (15,000 g, 1 h, 16 °C), and passed through a 0.45 µm filter before affinity chromatography. The protein was purified using an AKTA Start fast protein liquid chromatography system (GE Healthcare). Soluble inclusion bodies were loaded into a 5 ml HisTrap HP column (GE Healthcare) previously equilibrated with binding buffer (50 mM Tris, 0.5 M NaCl, 20 mM imidazole and 8 M urea, pH 8). The protein was eluted from the column with elution buffer (50 mM Tris, 0.5 M NaCl, 500 mM imidazole and 8 M urea, pH 8) and maintained under denaturing conditions.

**Apoplastic fluid extraction from the seeds.** The apoplastic fluid (AF) of the seeds was collected by a standard technique allowing the recovery of the components present in the intercellular spaces based on vacuum infiltration and centrifugation[45]. Briefly, imbibed seeds were carefully decoated and immersed in infiltration buffer (50 mM Tris-HCl, pH 7.5, and 0.6% NaCl). Five vacuum pulses of 10 s (separated by 30 s intervals) were applied using a vacuum pump. Infiltrated seeds were recovered, dried on a filter paper, placed in plastic syringe barrels inside centrifuge tubes and centrifuged for 20 min at 400 g at 4 °C. The AF was recovered

and stored at 4 °C until use at various concentration ratios (0, 1:2, 1:5, 1:10, 1:20 and 1:50) in TasA polymerization assays.

**TasA polymerization assays.** The kinetics of polymerization of TasA were monitored using thioflavin T (ThT) and detected by fluorescence[22]. The assay was carried out in a 96-well microtitre plate. Briefly, purified TasA protein was diluted in buffer (20 mM Tris and 50 mM NaCl, pH 7) or AF to final concentrations of 0.4 mg ml⁻¹ or 0.2 mg ml⁻¹. ThT solution was added to the wells to a final ThT concentration of 20 μM in a final volume of 100 μl. The fluorescence signal was measured in a fluorescence microplate spectrophotometer reader (438 nm excitation and 495 nm emission, with a 475 nm cut-off) at 30 °C under shaking. Each protein concentration was assayed in triplicate.

**Infection assays in the plants.** Assays of *B. cinerea* infection were carried out in 5–6-week-old plants. Fungal conidia were collected from light-grown culture in sterile distilled water and filtered through a 40 μm cell strainer to remove remaining hyphae. For inoculation, the conidial suspension was adjusted to 10⁵ conidia per ml in half-strength filtered (0.45 μm) grape juice (100% pure organic). Each leaf was inoculated with 5 μl droplets of conidial suspension. The pots were covered with a plastic dome and placed in a growth chamber. The leaves were imaged 72 h after the inoculation and the size of the lesions was determined using ImageJ software. The size of the lesions in experimental plants was normalized to a medium lesion size in the control plants, which was assumed as 100% disease percentage.

**Purification of oil bodies from the seeds.** Oil bodies were purified as previously described[46], with some modifications according to ref. [47]. Briefly, the seeds were soaked in 0.1 M sodium bicarbonate at pH 9.5 (adjusted with 0.1 M NaOH) for 2 h and soaking medium was discarded. The soaked seeds were ground in the same medium with a mortar and a pestle. The slurry was transferred to 2 ml tubes and centrifuged at 10,000 g for 30 min at 4 °C. The upper layer was transferred to a clean tube, dispersed in washing solution (7 M urea, 1:4 w/w) and centrifuged (10,000 g, 30 min). To remove residue of washing solution, the fat pad was isolated and suspended in water (1:4 w/w) and centrifuged (10,000 g, 30 min). After centrifugation, the creamy layer was collected and stored at 4 °C. Analyses were completed within 24 h.

**Pulldown assay.** To study possible TasA interactions with the proteins present in the seeds, we carried out a pulldown assay using a nickel-loaded affinity resin (Protino Ni-TED 2000, Macherey-Nagel); purified TasA was used as a bait protein for protein extract obtained from imbibed seeds. Imbibed and decoated seeds were ground to a fine powder using a mortar and a pestle in liquid nitrogen. The powder was transferred to a clean 15 ml tube and resuspended in extraction buffer (50 mM Tris-HCl, pH 7.5, 150 mM NaCl, 10 mM EDTA, 1% Triton X-100, 10% glycerol, 10 mM dithiothreitol (DTT), 1 mM phenylmethylsulfonyl fluoride, 1 mM NaF and 1% protease inhibitor cocktail). The samples were homogenized using a vortex, sonicated by 5 pulses of 20 s at 80% amplitude and centrifuged for 5 min at 9,000 g at 4 °C. The supernatant was transferred to a new tube and used immediately. Purified TasA was added to the seed protein extract to a final concentration of 50 μM and incubated at 4 °C overnight. Then, the mixture was incubated with equilibrated resin at 4 °C overnight. Control resin that did not contain bait protein was added to the seed protein extract. Flow-through, washing and elution fractions were collected and subjected to trichloroacetic acid protein precipitation. Precipitated proteins were resuspended in 1x Laemmli sample buffer (Bio-Rad) and heated at 100 °C for 5 min. The proteins were separated via SDS–PAGE through 12% polyacrylamide gels.

**Mass spectrometry analysis of protein bands.** The protein bands of interest were identified by high-performance liquid chromatography/electrospray ionization tandem mass spectrometry (HPLC-ESI-MS/MS). Briefly, the bands were cut out after electrophoresis, washed and destained. Subsequently, the disulfide bridges were reduced with DTT and cysteines were alkylated by iodoacetamide; in-gel trypsin digestion was performed to extract peptides from the protein samples. The entire process was carried out automatically using an automatic digester (DigestPro, Intavis Bioanalytical Instruments). The peptides were concentrated and desalted using a C18 ZORBAX 300SB-C18 capture column (Agilent Technologies; 5×0.3 mm, 5 μm particle diameter and 300 Å pore size) in a gradient of 98% H₂O:2% acetonitrile (ACN)/0.1% formic acid (FA) at a flow rate of 20 μl min⁻¹ for 6 min. The capture column was connected through a 6-port valve to a ZORBAX 300SB-C18 analytical column (Agilent Technologies; 150×0.075 mm, 3.5 μm particle diameter and 300 Å pore size). Elution of the samples from the capture column was performed by a gradient of 0.1% FA in water as a mobile phase A and 0.1% FA in 80% ACN/20% water as a mobile phase B. The LC system was coupled to a nanospray source (CaptiveSpray, Bruker Daltonics) and a three-dimensional (3D) ion trap mass spectrometer (amaZon speed ETD, Bruker Daltonics) operating in the positive mode at a capillary voltage of 1,500 V and a sweep range of *m/z* 300–1,500. Data-dependent acquisition was carried out in the automatic mode with sequential collection of the full scan MS spectra (*m/z* 300–1,400), followed by collection of MS spectra in tandem via collision-induced dissociation of the 8 most abundant ions. ProteinScape 3 software (Bruker Daltonics) coupled to Mascot 3.1 search engine (Matrix Science) was used for identification by matching the MS/MS data against the Cucurbit genomes databases[48].

**Far-western blotting.** The interaction between TasA and oleosin was examined by far-western blotting[49]. Purified proteins were separated by SDS–PAGE through a 15% acrylamide gel and electrophoretically transferred onto PVDF membranes using a Trans-Blot Turbo transfer system (Bio-Rad) and PVDF transfer packs (Bio-Rad). The proteins were renatured by incubation of the membrane in denaturing and renaturing buffer (100 mM NaCl, 20 mM Tris, pH 7.6, 0.5 mM EDTA, 10% glycerol, 0.1% Tween 20, 2% skim milk powder and 1 mM DTT) containing decreasing concentrations of guanidine HCl (6, 3, 1, 0.1 and 0 M) and the membrane was blocked with blocking buffer (TBS containing 0.1% Tween 20 and 5% skim milk). Then, the membrane was incubated in protein-binding buffer (100 mM NaCl, 20 mM Tris, pH 7.6, 0.5 mM EDTA, 10% glycerol, 0.1% Tween 20, 3% skim milk powder and 1 mM DTT) containing 10 μg of purified TasA at 4 °C overnight. Excess protein was removed, and bound TasA was subjected to immunodetection. The membrane was probed with an anti-TasA antibody (rabbit) at a 1:20,000 dilution in blocking buffer. A secondary anti-rabbit IgG antibody conjugated to horseradish peroxidase (Bio-Rad) was used at a 1:3,000 dilution in the same buffer. The membranes were developed using Pierce ECL western blotting substrate (Thermo Fisher).

**Gene expression analysis by RT–qPCR.** Total RNA was extracted from the frozen tissues collected at various timepoints after the treatments by phenol/chloroform extraction followed by precipitation with 1.5 M LiCl[50] and sodium acetate[51]. For quantitative PCR with reverse transcription (RT–qPCR) assays, the RNA concentration was adjusted to 100 ng μl⁻¹. Then, 1 μg of DNA-free total RNA was reverse transcribed into complementary DNA using SuperScript III reverse transcriptase (Invitrogen) and random primers in a final reaction volume of 20 μl according to the manufacturer's instructions. RT–qPCR was performed using an iCycler-iQ system and an iQ SYBR Green Supermix kit from Bio-Rad according to the manufacturer's instructions. The RT–qPCR cycle included the following steps: 95 °C for 3 min, followed by PCR amplification using a 40-cycle amplification programme (95 °C for 20 s, 56 °C for 30 s and 72 °C for 30 s), followed by a third step at 95 °C for 30 s. To evaluate the melting curve, 40 additional cycles were performed for 15 s starting at 75 °C, with stepwise temperature increases of 0.5 °C per cycle. The *act7* gene was used as a reference for data normalization. The target genes were amplified using the primers listed in Supplementary Table 2. Primer efficiency tests and confirmation of the specificity of the amplification reactions were performed as previously described[52]. Relative transcript abundance was estimated using the ΔΔ cycle threshold (Ct) method[53]. Transcriptional data of the target genes were normalized to the *act7* gene and are shown as the fold change in the expression levels of the target genes in each experimental treatment compared to those in the control treatment. The relative expression ratios were calculated as the differences between the qPCR threshold cycles (Ct) of the target gene and the Ct of the *act7* gene ($\Delta Ct = Ct_{\text{gene of interest}} - Ct_{act7}$). The fold-change values were calculated as $2^{-\Delta\Delta Ct}$, assuming that a single PCR cycle represents a two-fold difference in the template abundance[54,55]. RT–qPCR analyses were performed in technical triplicates using three independently isolated RNA samples (biological triplicates).

**Total transcriptome analysis.** For RNA sequencing analysis, 100 bp single-end read libraries were prepared using a TruSeq stranded total RNA kit (Illumina). The libraries were sequenced using a NextSeq550 instrument (Illumina). Raw reads were preprocessed by NextSeq System Suite v.2.2.0. using specific NGS technology configuration parameters[56]. This preprocessing removes low-quality, ambiguous and low-complexity stretches, linkers, adapters, vector fragments and contaminated sequences, and preserves the longest informative parts of the reads. SeqTrimNext also discarded sequences shorter than 25 bp. Subsequently, clean reads of the BAM files were aligned and annotated by Bowtie2[57] using the *Cucumis melo* genome v.4.0[58] as the reference; these data were then sorted and indexed using SAMtools v.1.484110[59]. Uniquely localized reads were used to calculate the read number values for each gene by Sam2counts (https://github.com/vsbuffalo/sam2counts). Differentially expressed genes (DEGs) in the treatment samples were analysed by DEgenes Hunter[60], which provides a combined *P* value calculated on the basis of the Fisher method[61] using nominal *P* values provided by edgeR[62] and DEseq2[63]. This combined *P* value was adjusted by the Benjamini-Hochberg procedure[48] (false discovery rate approach) and used to rank all obtained DEGs. For each gene, a combined *P* value < 0.05 and log₂ fold change (FC) > 1 or < −1 were considered significance thresholds. A heatmap and DEG clustering were generated using ComplexHeatmap in R Studio and Kobas 2.0[64]. Only profiles with a *P* value < 0.05 were considered in the present study. DEGs annotated using the *Cucumis melo* genome were processed to identify the Gene Ontology functional categories using sma3s[65] and TopGo software[66]. Kyoto Encyclopedia of Genes and Genomes (KEGG) pathways and enrichment were estimated using Bioconductor packages (Bioconductor.org) GGplot2, ClusterProfiler, DOSE and EnrichPlot in R Studio. The data were deposited in the GEO database under the reference GSE175611.

**CLSM.** Bacteria, ROS and OBs inside the seeds were visualized by CLSM. Seeds were collected at the specified hours after treatment and were transversally cut with a scalpel. A drop of glycerol was applied to the sections, which were placed into 1.5-mm-thick cover glasses ($22 \times 22$ mm). The bacteria were stained during inoculum preparation by CellTracker CM-DiI dye or in situ by Hoechst solution. Oil bodies and ROS were stained with Nile red (1:1,000 v/v) or dihydrorhodamine 123 (1:500 v/v), respectively. The images were acquired using a Leica SP5 confocal microscope with a ×HCX IRAPO L 25.0×0.95 WATER objective. To image the purified OB suspension, a drop of Nile red and Fast Green FCF-stained preparation was applied onto a patch of polymerized 1% agarose on a glass slide, which was covered with a coverslip. The images were acquired using a Leica SP5 confocal microscope with HCX PL APO lambda blue 63.0×1.40 OIL UV objective. Image processing was performed using Leica Application Suite Advance Fluorescence v.2.7.3.9723 (LCS Lite, Leica Microsystems) and FIJI/ImageJ software. Laser settings, scan speed, photomultiplier detector gain and pinhole aperture were constant for all acquired image stacks for each experiment.

**TEM.** For TEM analysis, 16 h after treatment, seeds were fixed directly using 2% paraformaldehyde, 2.5% glutaraldehyde and 0.2 M sucrose mixture in 0.1 M phosphate buffer (PB) overnight at 4 °C. After three washes in PB, the portions were excised from the micropylar endosperm region and postfixed in 1% osmium tetroxide solution in PB for 90 min at room temperature, followed by PB washes and 15 min of stepwise dehydration in an ethanol series (30%, 50%, 70%, 90% and 100% twice). Between the 50% and 70% steps, the samples were incubated in-bloc in 2% uranyl acetate solution in 50% ethanol at 4 °C overnight. After dehydration, the samples were gradually embedded in low-viscosity Spurr's resin (resin:ethanol, 1:1, 4 h; resin:ethanol, 3:1, 4 h; and pure resin overnight). The sample blocks were embedded in capsule moulds containing pure resin for 72 h at 70 °C. To image purified OBs in suspension or TasA protein under various polymerization conditions, carbon-coated copper grids were deposited over the sample drops and incubated at room temperature for 2 h. After incubation, the grids were washed in phosphate-buffered saline for 5 min, negatively stained with uranyl acetate (1%) for 20 s and washed once with water for 30 s.

For immunolabelling assays, carbon-coated copper grids were deposited over the samples of the corresponding OB treatments. After 2 h of incubation, the grids were washed in phosphate-buffered saline for 5 min and blocked with Pierce protein-free (TBS) blocking buffer (Thermo Fisher) for 30 min. An anti-TasA primary antibody was used at a 1:150 dilution in blocking buffer, and grids were deposited over the drops of antibody solution and incubated for 1 h at room temperature. The samples were washed three times with TBS-T (50 mM Tris-HCl, 150 mM NaCl, pH 7.5, and 0.1% Tween 20) for 5 min and then exposed to a 10-nm-diameter immunogold-conjugated secondary antibody (20 nm goat anti-rabbit conjugate, BBI solutions) for 1 h at a 1:50 dilution. The samples were then washed twice with TBS-T and once with water for 5 min each time. Finally, the grids were treated with 2% glutaraldehyde for 10 min, washed in water for 5 min, negatively stained with 1% uranyl acetate for 20 s and washed once with water for 30 s. The samples were left to dry and imaged under an FEI Tecnai G2 20 TWIN transmission electron microscope at an accelerating voltage of 80 kV. The images were acquired using TIA FEI Imaging Software v.4.14.

**Metabolite extraction from plant tissues.** Metabolites from the samples of adult melon plants or radicles (MSV000084674, MSV000084278 and MSV000086360) were extracted by adding 1 ml methanol and vigorous vortexing. At least 2 or 3 replicates were done for each experiment, except for adult melon plants where only 1 replicate was done due to the high number of sections analysed for each plant. Due to the large sample number, weighting and normalization by biomass was not feasible. However, the size of plant sections was kept constant. To enable relative quantitative comparison between samples, we normalized peak areas to total ion count, which is a robust and widely applied strategy in non-targeted metabolomics. In addition, sulfamethazine (1 μM) was added to all samples as an internal standard and was used for normalization. After incubation for 2 h, the extracts were centrifuged at 14,000 g for 30 min. The acquired supernatants were stored at −80 °C until use in liquid chromatography–tandem mass spectrometry (LC–MS/MS).

Metabolites from seeds and seedlings (MSV000088139) were extracted after two rounds of methanol incubation with the powder obtained after grinding the tissue in liquid nitrogen and freeze-drying the samples. Three replicates were done for each condition. The samples were transferred to weighted vials and the methanol was evaporated at room temperature. Extracts were then resuspended in 1 ml methanol and centrifuged at 14,000 g for 20 min at 4 °C. Vials were weighed after the extractions to adjust the final concentrations to 1 mg ml⁻¹, thus no internal standard was used.

In both cases, extraction controls were performed by adding 3 samples with pure methanol used for the extraction as a blank. Differences in abundance of metabolites were estimated on the basis of relative comparisons between equally treated samples due to the absence of authentic standards in the non-targeted metabolomics approach.

**LC–MS/MS.** Non-targeted LC–MS/MS analysis was performed using a Q-Exactive Quadrupole-Orbitrap mass spectrometer coupled to a Vanquish ultra high-performance liquid chromatography (UHPLC) system (Thermo Fisher) (MSV000086360, MSV000084674 and MSV000084278) or a Shimadzu Nexera X2 UHPLC system, with attached photodiode array detector coupled to a Shimadzu 9030 QTOF mass spectrometer (MSV000088139).

For the Q-Exactive instrument, 5 μl of the samples were injected for separation on a UHPLC with a C18 core-shell column (Kinetex, $50 \times 2$ mm, 1.8 μm particle size, 100 Å pore size, Phenomenex). For the mobile phase, we used a flow rate of 0.5 ml min⁻¹ (solvent A: $H_2O$ + 0.1% FA; solvent B: acetonitrile + 0.1% FA). During the chromatographic separation, we applied a linear gradient from 0–0.5 min 5% B, 0.5–4 min 5–50% B, 4–5 min 50–99% B, followed by a 2 min washout phase at 99% B and a 2 min re-equilibration phase at 5% B. For positive mode MS/MS acquisition, the electrospray ionization was set to a 35 a.u. sheath gas flow, 10 a.u. auxiliary gas flow, 2 a.u. sweep gas flow and 400 °C auxiliary gas temperature. The spray voltage was set to 3.5 kV with an inlet capillary of 250 °C. The S-lens voltage was set to 50 V. MS/MS product ion spectra were acquired in data-dependent acquisition mode. MS1 survey scans (150–1,500 $m/z$) and up to 5 MS/MS scans per data-dependent acquisition duty cycle were measured with a resolution of 17,500. The C-trap fill time was set to a maximum of 100 ms or until the Automatic Gain Control target of $5 \times 10^5$ ions was reached. The quadrupole precursor selection width was set to 1 $m/z$. Normalized collision energy was applied stepwise at 20, 30 and 40%, with $z = 1$ as default charge state. MS/MS scans were triggered with apex mode within 2–15 s from their first occurrence in a survey scan. Dynamic precursor exclusion was set to 5 s. Precursor ions with unassigned charge states and isotope peaks were excluded from MS/MS acquisition.

For the QTOF instrument, 2 μl were injected into a Waters Acquity Peptide BEH C18 column (1.7 μm, 300 Å, $2.1 \times 100$ mm). The column was maintained at 40 °C and run at a flow rate of 0.5 ml min⁻¹, using 0.1% formic acid in $H_2O$ as solvent A and 0.1% formic acid in acetonitrile as solvent B. A gradient was employed for chromatographic separation starting at 5% B for 1 min, then 5% to 85% B for 9 min, 85% to 100% B for 1 min, and finally held at 100% B for 4 min. The column was re-equilibrated to 5% B for 3 min before the next run was started. The LC flow was switched to the waste for the first 0.5 min, then to the MS for 13.5 min, then back to the waste to the end of the run. The photodiode array detector acquisition was performed in the range 200–400 nm at 4.2 Hz, with 1.2 nm slit width. The flow cell was maintained at 40 °C. The MS system was tuned using standard NaI solution (Shimadzu). The same solution was used to calibrate the system before starting. System suitability was checked by including a standard sample made of 5 μg ml⁻¹ thiostrepton, which was analysed regularly in between batches of samples.

All samples were analysed in positive polarity, using data-dependent acquisition mode. In this regard, full scan MS spectra ($m/z$ 400–4,000, scan rate 20 Hz) were followed by 3 data-dependent MS/MS spectra ($m/z$ 400–4,000, scan rate 20 Hz) for the 3 most intense ions per scan. The ions were selected when they reached an intensity threshold of 1,000, isolated at the tuning file Q1 resolution, fragmented using collision-induced dissociation with collision energy ramp (CE 10–40 eV) and excluded for 0.05 s (1 MS scan) before being reselected for fragmentation. The parameters used for the ESI source were: interface voltage 4 kV, interface temperature 300 °C, nebulizing gas flow 3 l min⁻¹ and drying gas flow 10 l min⁻¹.

In both cases, a mixture of 10 mg ml⁻¹ each of sulfamethazine, sulfamethizole, sulfachloropyridazine, sulfadimethoxine, amitriptyline and coumarin was run after every 96 injections for quality control.

**Preprocessing for data analysis and MS/MS network analysis.** Raw spectra were converted to .mzXML files using MSconvert (ProteoWizard). Feature identification was performed by MZmine open source software[67,68] version 2.37 using the settings listed in Supplementary Table 3; this generated the .mgf files and quantification tables for statistical analyses via Metaboanalyst[69] and feature-based molecular network in the Global Natural Product Social Molecular Networking (GNPS) environment[70] using Ion Identity Network[71].

**Feature-based molecular networking and spectral library search.** Molecular networking and library search were performed using the GNPS environment with the settings listed in Supplementary Table 4 and visualized using Cytoscape software[72]. Putative annotation of detected features was obtained using automatic library search through the GNPS environment[73], network annotation propagation[74], chemical classification using ClassyFire[75] and MolNetEnhancer workflows[76].

Mirror plots were done using GNPS and https://metabolomics-usi.ucsd.edu/[77], comparing mzspec of the selected features and the metabolites recorded in MS/MS databases (Extended Data Fig. 10). Annotations were done according to guidelines in ref. [78] (Supplementary Table 8), and levels are indicated in each figure.

The automatic workflow for the analysis of the radicles from seeds bacterized with *Bacillus subtilis* subsp. *subtilis* NCBI 3610 and mutant strains dataset (MSV000084674) can be accessed at: https://massive.ucsd.edu/ProteoSAFe/dataset.jsp?task=b2e3636335d24c2da72634b7c6c8b63f; feature-based molecular networking: https://gnps.ucsd.edu/ProteoSAFe/status.jsp?task=811bd911de4f48a1b59510215559afa2; network annotation propagation: https://proteomics2.ucsd.edu/ProteoSAFe/status.jsp?task=5d2297ea755b4d3986cd92d8e297e603; and

MolNetEnhancer: https://proteomics2.ucsd.edu/ProteoSAFe/status.jsp?task=45823b975a8547e4b76b82ce8e037142.

Automatic workflow for the analysis of the radicles from the seeds treated with fengycin and TasA (MSV000084278) can be accessed at: https://massive.ucsd.edu/ProteoSAFe/dataset.jsp?task=fc05e1b986cb4073b56d9f9a044f0a2f; feature-based molecular networking: https://gnps.ucsd.edu/ProteoSAFe/status.jsp?task=446bcbde8940460b89eda8c0bddfdd27; network annotation propagation: https://gnps.ucsd.edu/ProteoSAFe/status.jsp?task=eb776d82cf5f438282514e5d438c851e; and MolNetEnhancer: https://gnps.ucsd.edu/ProteoSAFe/status.jsp?task=1f316080df4749a3ae00f2d644c95114.

Automatic workflow for the analysis of adult plants grown from control seeds or seeds bacterized with *Bacillus subtilis* subsp. *subtilis* NCBI 3610 (MSV000086360) can be accessed at the following addresses: https://massive.ucsd.edu/ProteoSAFe/dataset.jsp?task=1090c82889a04e4ba9be1058a8a5d6db; feature-based molecular networking: https://proteomics2.ucsd.edu/ProteoSAFe/status.jsp?task=63d6d0c95b3e4d8d846d376499bb544e; network annotation propagation: https://gnps.ucsd.edu/ProteoSAFe/status.jsp?task=914293e2bc3d4492bb9e0de3c7c49b2c; and MolNetEnhancer: https://gnps.ucsd.edu/ProteoSAFe/status.jsp?task=e72d0b1205444e77880befcc9d23d5a8.

Automatic workflow for the analysis of the seeds and seedlings after seed treatments with bacteria, fengycin or TasA (MSV000088139) at 0, 24 and 120 h after treatment can be accessed at: https://massive.ucsd.edu/ProteoSAFe/dataset.jsp?task=72bb762b7b8d45da8d3c5c9005a425d8; feature-based molecular networking: https://gnps.ucsd.edu/ProteoSAFe/status.jsp?task=c6cce7f2ec2048febbc6b53b09e69525;

network annotation propagation: https://proteomics2.ucsd.edu/ProteoSAFe/status.jsp?task=bee18fc3c2c44e70b9b313f92ef3402; and MolNetEnhancer: https://proteomics2.ucsd.edu/ProteoSAFe/status.jsp?task=eb669e7594bf440d88c581b34636e7c2.

**Proportionality score.** The proportionality score was calculated between two directly connected nodes across the entire molecular network using the following equation:

$$\text{Proportionality} = log\left(\frac{\text{Ns1/Ms1}}{\text{Ns2/Ms2}}\right),$$

where Ns1 and Ms1 correspond to the peak area of the detected features N and M in sample S1, while Ns2 and Ms2 correspond to the peak area of the detected features N and M in sample S2. A constant ($k = 1.0 \times 10^{-10}$) is added to each value to avoid including any zero values during the calculation. Chemical proportionality scores are available within the GNPS environment and can be accessed at: https://proteomics2.ucsd.edu/ProteoSAFe/status.jsp?task=7e8109b3ece945979f8b26ab7a48d615 (chemical proportionality table calculated between non-treated seeds at 0 and 24 h); https://proteomics2.ucsd.edu/ProteoSAFe/status.jsp?task=488cd1ca8e7540bb8033f79f0712fa3d (chemical proportionality table calculated between seeds treated with fengycin at 0 and 24 h); and https://proteomics2.ucsd.edu/ProteoSAFe/status.jsp?task=fca204a7579448bc9f821d08863fc0f9 (chemical proportionality table calculated between seeds treated with TasA at 0 and 24 h).

**Collection of plant material for 3D mass spectral molecular cartography.** Individual parts of melon plants grown from control and bacterized seeds were sampled. Plants were sectioned in the same number of parts to make results comparative. For this, roots, stems and leaves were numbered, and coordinates were taken for later 3D representation and visualization. Approximately 200 mg of fresh individual plant parts was collected in 1.5 ml microcentrifuge tubes and flash frozen under liquid nitrogen. The samples were stored at −80 °C until further analysis.

**3D modelling and visualization.** Melon plants from non-bacterized seeds were used as the model plant for 3D representation of metabolomic features due to non-significant changes in morphology between plants. Approximately 250 images of melon plants taken from different angles and perspectives were converted into high-definition 3D meshes using Autodesk ReCap. These 3D meshes were then exported to the .obj format and edited using Meshmixer. Point coordinates of sampled plant parts were added using Meshlab[79] and the 3D models were exported to the .stl format. Representative samples of roots, stems and leaves were obtained. Point coordinates and LC–MS data were combined into a .csv file and mapped on the 3D models using Ili[80].

**Statistics and reproducibility.** Statistical significance was assessed by appropriate tests (see figure legends). All analyses were performed using GraphPad Prism version 6. *P* values < 0.05 were considered significant. Asterisks indicate the level of statistical significance: *$P < 0.05$, **$P < 0.01$, ***$P < 0.001$ and ****$P < 0.0001$. Statistical analysis of metabolomic datasets was performed by Metaboanalyst v.5.0 (ref. [69]), after data filtering by interquartile range (IQR). When needed, data were normalized to the feature corresponding to the internal standard of the samples. Experiments were repeated at least three times independently with similar results.

**Reporting summary.** Further information on research design is available in the Nature Research Reporting Summary linked to this article.

## Data availability
The RNA-seq data are deposited in the GEO database under the reference GSE175611. Metabolomics data are deposited at https://massive.ucsd.edu/ with the identifiers MSV000084674 (data from radicles from seeds bacterized with *Bacillus subtilis* subsp. *subtilis* NCBI 3610 and mutant strains), MSV000084278 (data from radicles from seeds treated with fengycin and TasA), MSV000086360 (data from adult plants grown from control seeds or seeds bacterized with *Bacillus subtilis* subsp. *subtilis* NCBI 3610) and MSV000088139 (data from seeds and seedlings after seed treatments with bacteria, fengycin or TasA). Source data are provided with this paper.

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

## Acknowledgements

We thank S. M. Rojas for technical support, J. G. Maldonado from the Ultrasequencing Unit of the SCBI-UMA for RNA sequencing, C. Cárdenas for protein analysis (Proteomic Unit, SCAI-UMA), J. F. L. Téllez for technical support on TEM analysis, J. Pearson for technical support on confocal laser scanning microscopy analysis (Bionand), S. Elsayed (Leiden University) for technical support on metabolomics analysis and T. Ketelaar (Wageningen University and Research) for suggestions on the interpretation of TEM

images. This work was supported by grants from ERC Starting Grant (BacBio 637971) Plan Nacional de I+D+i of Ministerio de Economía y Competitividad and Ministerio de Ciencia e Innovación (AGL2016-78662-R and PID2019-107724GB-I00), Proyectos I+D+I en el marco del Programa Operativo FEDER Andalucía (UMA18-FEDERJA-055) and Ayudas a Proyectos I+D+I Programa PAIDI 2020 of Junta de Andalucía (P20_00479). M.V.B.-C. received an FPU contract (FPU17/03874) from the Ministerio de Ciencia, Innovación y Universidades. C.M.-S. was funded by the programme Juan de la Cierva Incorporación (IJC2018-036923-I). D.P. was supported by the German Research Foundation (DFG) with Grant PE 2600/1. A.M.C.-R. and P.C.D. were supported by the National Institutes of Health (NIH) grant DP2GM137413. P.C.D. was supported by the Gordon and Betty Moore Foundation through Grant GBMF7622, and the U.S. National Institutes of Health for the Center (P41 GM103484, R03 CA211211, R01 GM107550).

## Author contributions

D.R. conceived and designed the work, drafted and edited the text; M.V.B.-C. collected most of the experimental data, and drafted the manuscript; C.M.-S. designed, collected and analysed MS data, and edited the manuscript; A.M.C.-R. collected and analysed MS data, and edited the text; D.P. collected MS data and edited the text; L.D.-M. informatically analysed data and drafted figures; V.J.C. supervised MS analyses and edited the text; A.d.V. substantially revised and edited the text; A.P.-G. substantially revised and edited the text; P.C.D. substantially revised and edited the text.

## Competing interests

The authors declare no competing interests.

## Additional information

**Extended data** is available for this paper at https://doi.org/10.1038/s41564-022-01134-8.

**Correspondence and requests for materials** should be addressed to D. Romero.

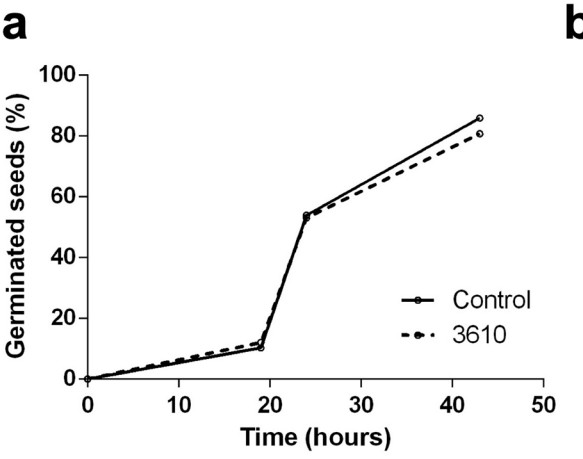

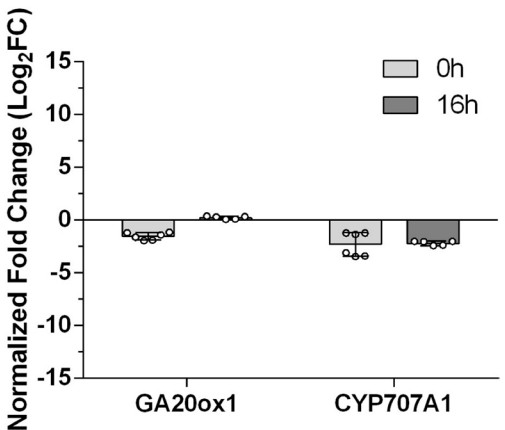

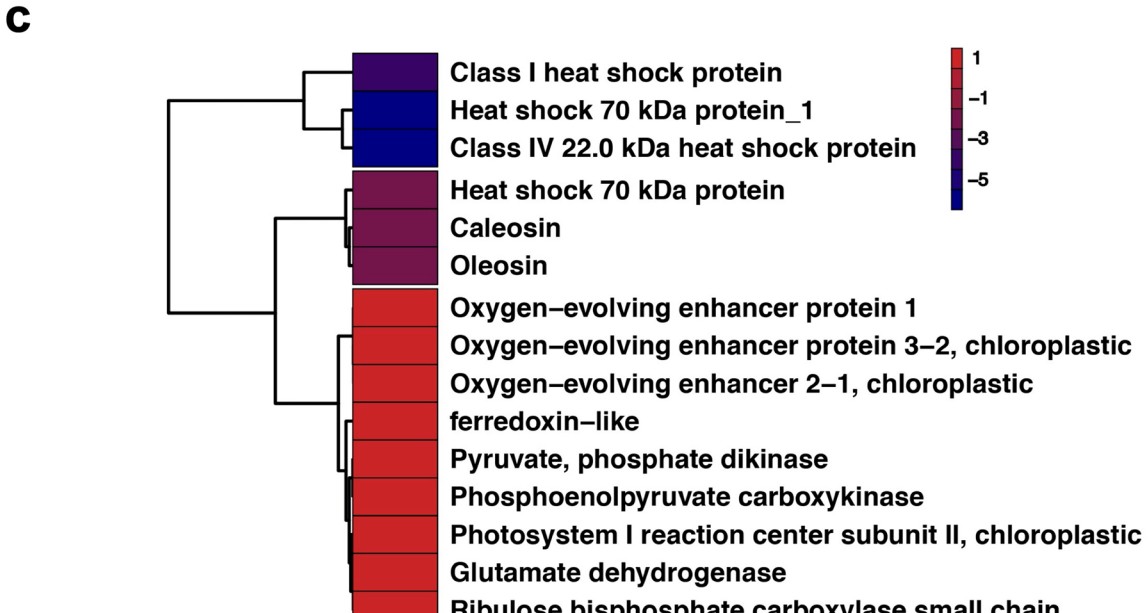

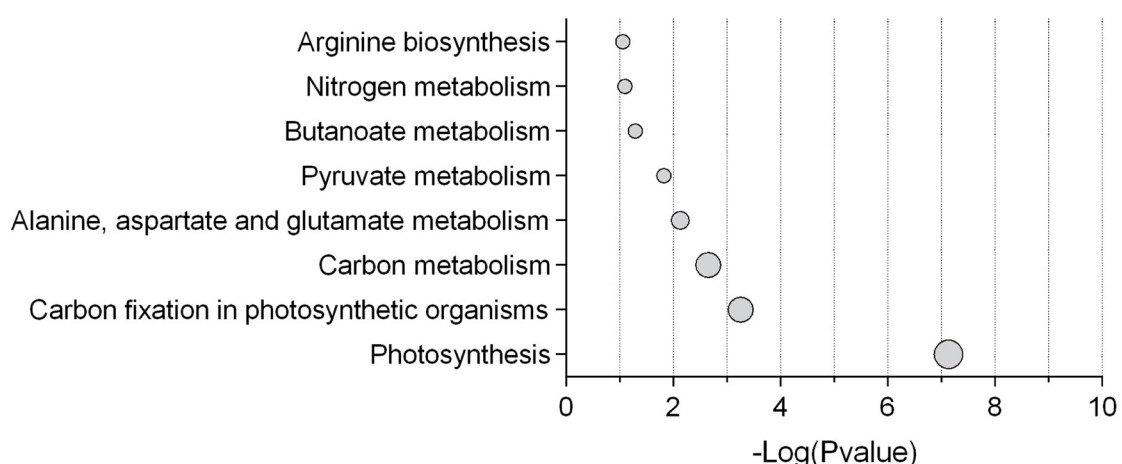

**Extended Data Fig. 1 | See next page for caption.**

**Extended Data Fig. 1 | Seed treatment with *B. subtilis* cells does not induce an increase in the germination rate but accelerates post germination progression. a** Germination rates of untreated and bacterized seeds represented as the percentage of germinated seeds versus time. **b** Relative expression levels of the GA20ox1 and CYP707A1 genes in bacterized seeds 0 and 16 hours after the treatment compared with those in the untreated seeds. The average values of two biological replicates with three technical replicates are shown with error bars representing SD. **c** Heatmap of fold changes of selected DEGs identified by total transcriptomic analysis of bacterized and control seeds (untreated) 16 hours after the treatment. Colour scale indicates the Log2(FC) of each DEG. **d** KEGG pathway enrichment analysis of all DEGs identified by RNA-seq of bacterized and control seeds 16 hours after the treatment.

**a**

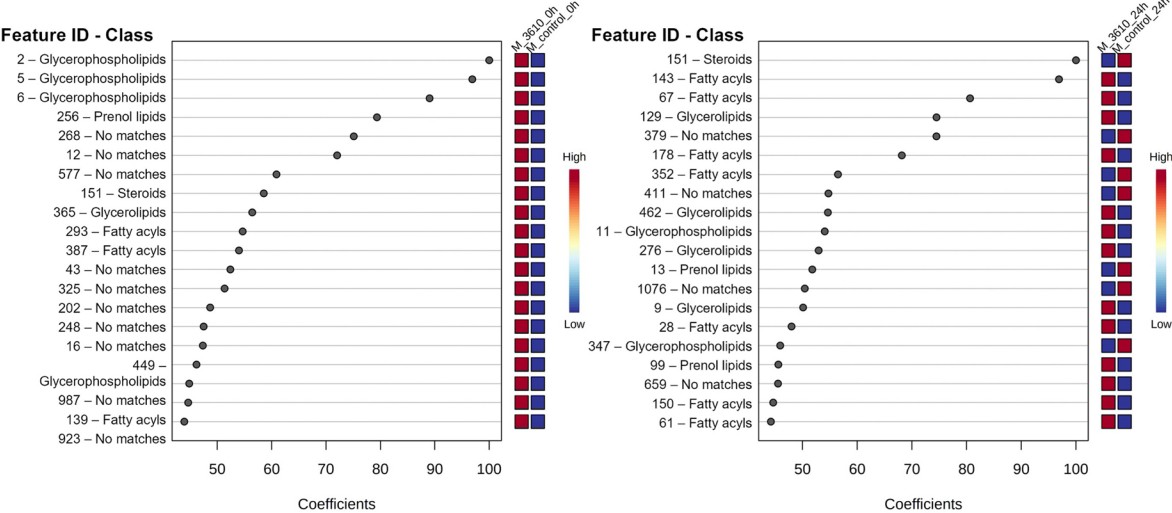

**b**

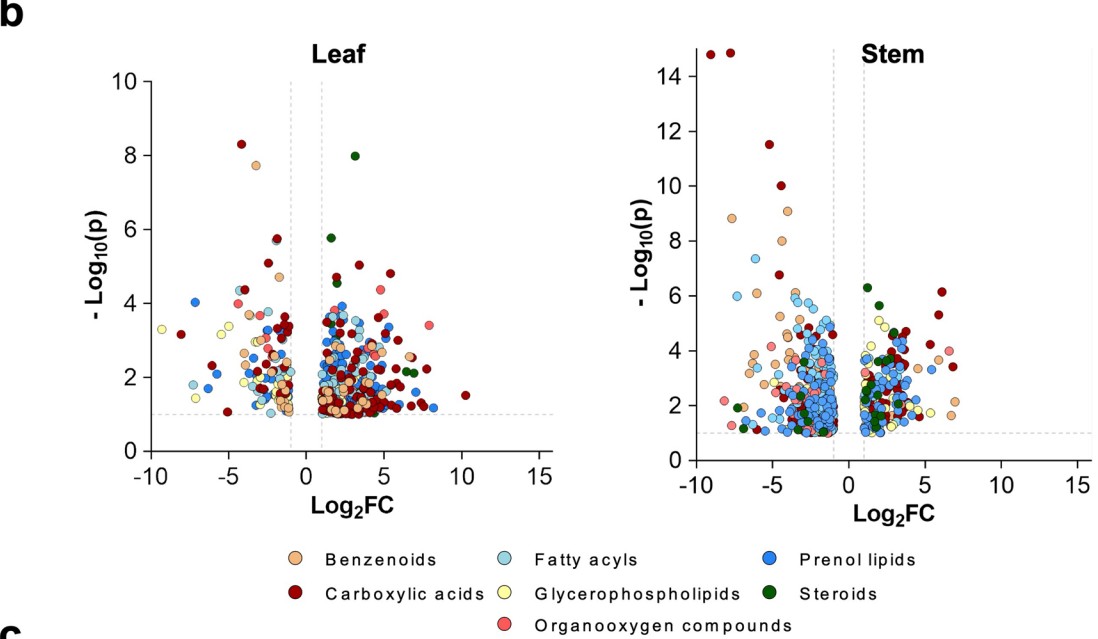

**c**

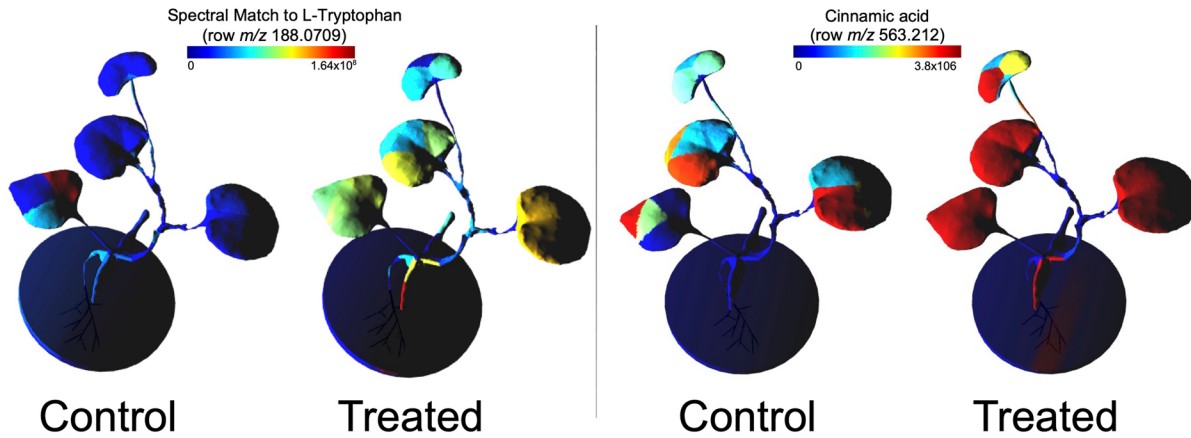

Control          Treated          Control          Treated

**Extended Data Fig. 2 | See next page for caption.**

**Extended Data Fig. 2 | Carboxylic acids and lipids represent the major metabolic changes in bacterized seeds. a** Top 20 features with high median weighted sum of absolute regression coefficient scores determined by PLS-DA and calculated using MetaboAnalyst, selected as features discriminating bacterized seeds from control seeds 0 hours after the treatment (left) or 24 hours after the treatment (right). Feature IDs are accompanied by chemical class according to Classyfire classification. Features in bold are selected as features discriminating any growth-promoting treatment from the control treatment. **b** Volcano plots showing the metabolites significatively increased or decreased in leaves (left) and stem (right) of plants emerged from bacterized seeds respecting control plants. Colours indicate the chemical class of each metabolite according to Classyfire. **c** Distribution of features identified by spectral match as L-tryptophan and cinnamic acid in 3D models representing the plants grown from the control or *B. subtilis*-treated seeds.

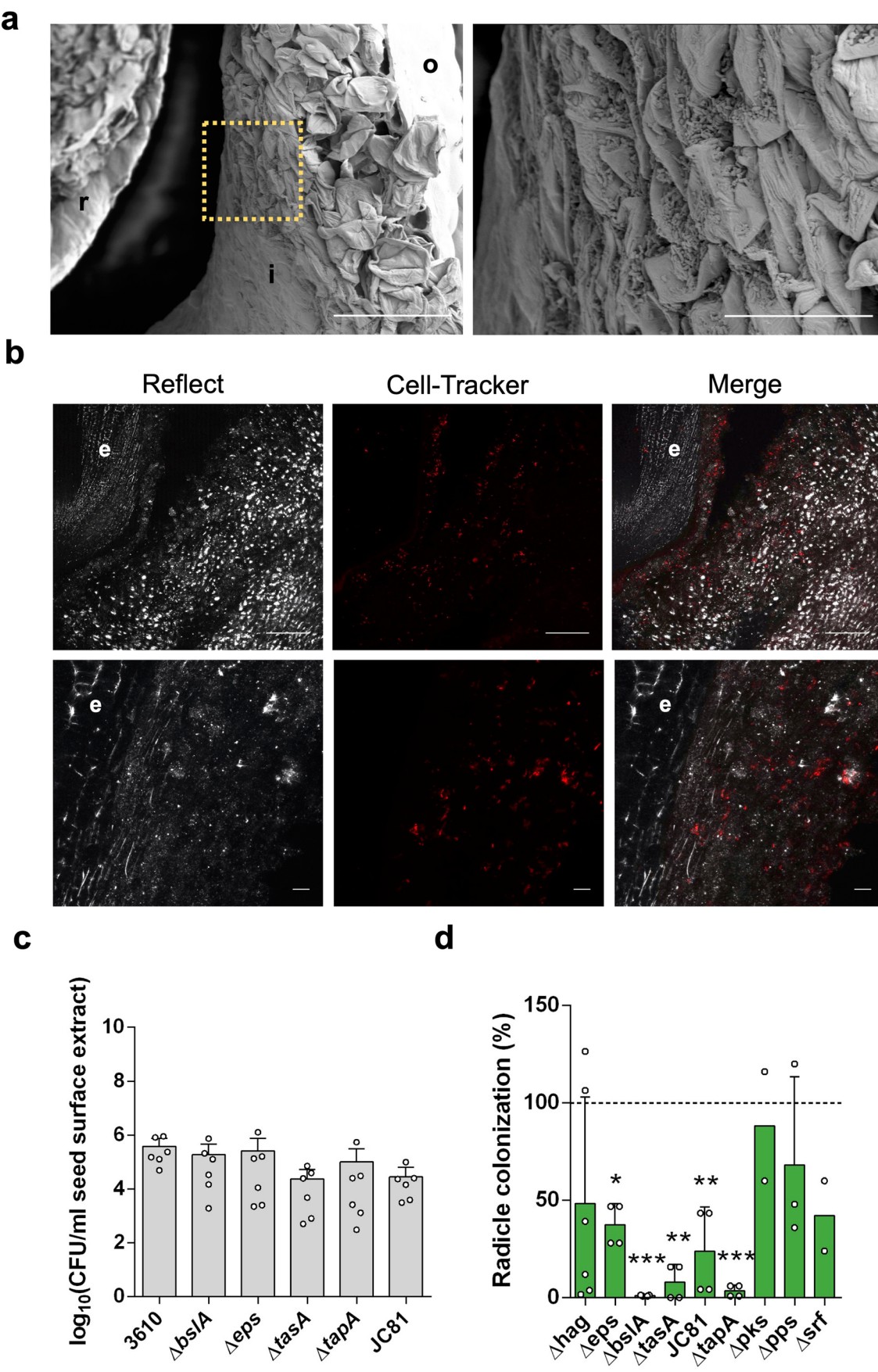

**Extended Data Fig. 3 | See next page for caption.**

**Extended Data Fig. 3 |** *B. subtilis* **cells enter the seeds and colonize the inner regions. a** Scanning electron microscopy micrographs of bacterized seeds 5 days after the treatment shows *B. subtilis* cells colonizing the inner side of the seed coat. Scale bars: 100 = µm (left image) and 30 µm (right image); r: radicle, i: inner seed coat, o- outer seed coat. **b** Confocal microscopy images of transversally cut bacterized seeds 16 hours after the treatment with fluorescently labeled *B. subtilis* cells. Scale bars: 100 µm (upper panels) or 10 µm (lower panels); e: embryo. **c** CFU counts of the surface extracts of the seeds bacterized with the WT (3610) strain and ECM mutants 1 hour after the treatment. Average values are shown with error bars representing SD ($n = 6$). **d** Radicle colonization of the ECM mutants relative to that of the WT assumed as 100% (discontinued line) in radicle extracts five days after seed treatment. Average values are shown. Error bars represent SEM. Statistical significance was assessed by one-way ANOVA with *post hoc* Dunnett's multiple comparisons test (each treatment vs. WT treatment except for Δsrf and Δpks) ($n = 3$ in WT, $n = 4$ in all ECM mutants except for Δsrf and Δpks, with $n = 2$; Δeps *$P = 0.0391$, ΔbslA ***$P = 0.0007$, ΔtasA **$P = 0.0014$, JC81 **$P = 0.0091$, ΔtapA ***$P = 0.0009$).

**a**

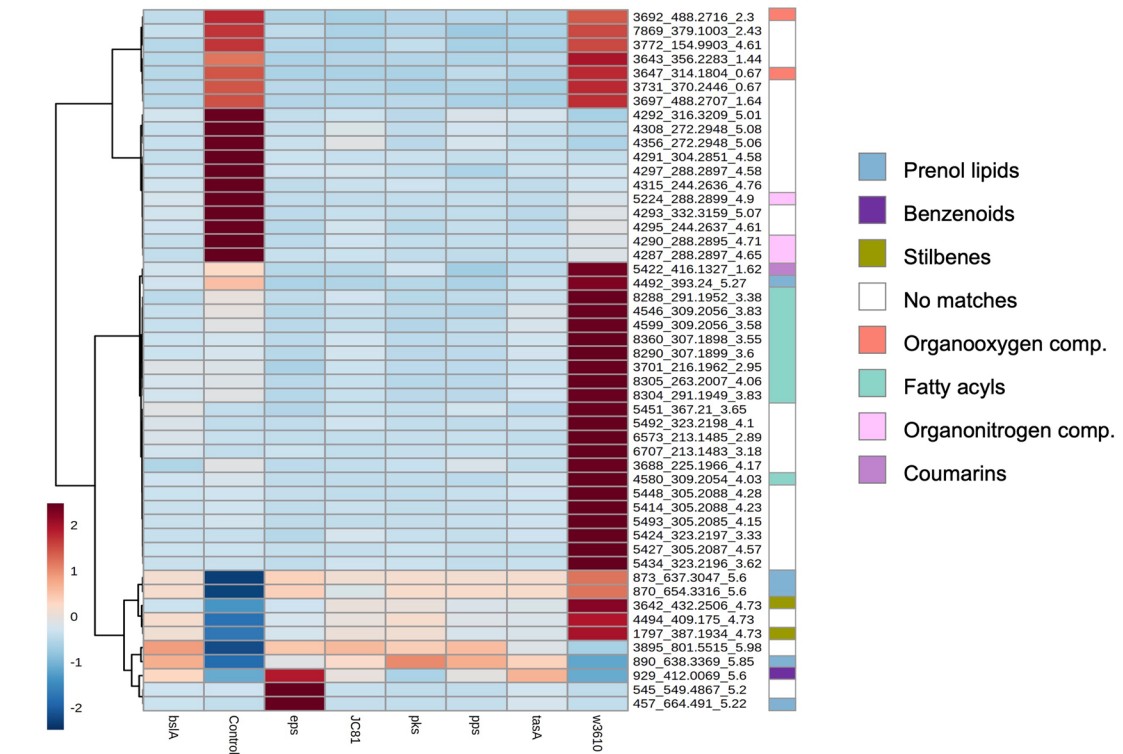

**b**

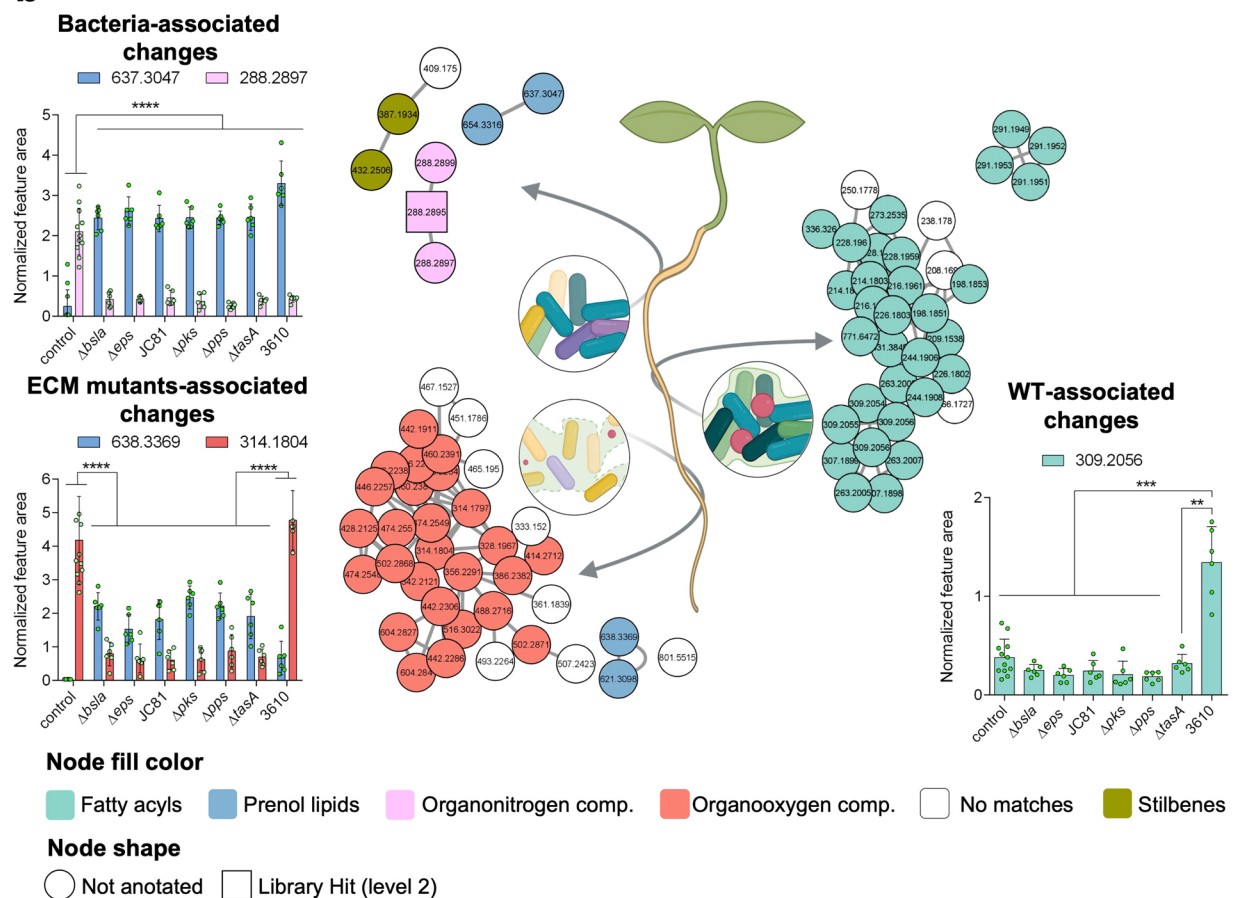

**Extended Data Fig. 4 | See next page for caption.**

**Extended Data Fig. 4 | Components of the extracellular matrix of *B. subtilis* trigger metabolic reprogramming of the seed radicles related to plant growth stimulation. a** Heatmap of the hierarchical clustering of the top 50 features of impacted molecular families in the radicles from bacterized seeds. The color code inside the heatmap depicts the relative fold change of each metabolite between groups. Color code accompanying feature names indicates their chemical class according to Classyfire. **b** Network analysis of representative features related to the presence of bacteria or ECM. Normalized abundances in the radicles in seeds subjected to various treatments are represented in features of all groups. Average values are shown with error bars representing SD. Statistical significance was assessed by one-way ANOVA with *post hoc* Dunnett's multiple comparisons test ($n = 6$ except in control, where $n = 12$; $p < 0.0001$). Created with BioRender.com.

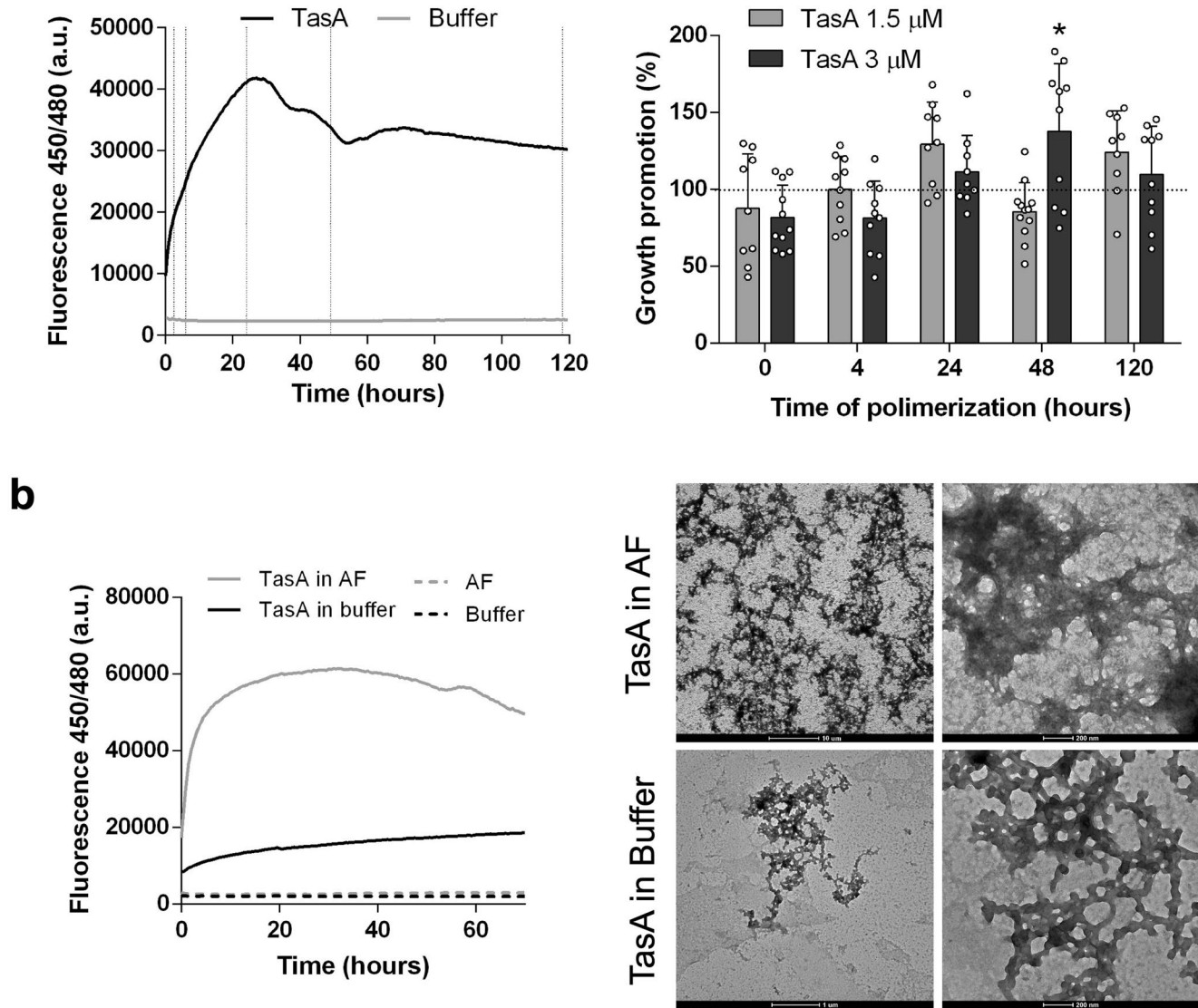

**Extended Data Fig. 5 | Aggregated forms of TasA are preferred under the apoplast conditions and have the highest radicle growth-promoting activity after seed treatments. a** Left: Polimeryzation dynamics of TasA in the fluorescence emission assays using ThT. Discontinued lines indicate time points when fractions were used for seed treatments. Right: Percentage of radicle growth promotion from the seeds treated with TasA at various polymerization states normalized to the radicle area of the control samples (100%, discontinued line) five days after seed treatment. Average values are shown. Error bars represent SD. Statistical significance was assessed by after one-way ANOVA with *post hoc* Dunnett's multiple comparisons test (each treatment vs. control treatment) ($n = 11$, 12, 10, 11 and 9 for control, $n = 9$, 10, 9, 12 and 9 for TasA 1.5 μM and $n = 11$, 10, 9, 10 and 10 for TasA 3 μM, referring to 0, 4, 24, 48 and 120 hours respectively). **b** Left: Fluorescence emission of ThT over time in the TasA polymerization assay in polymerization buffer (buffer) or in apoplastic fluid (AF). Right: Representative transmission electron microscopy micrographs of TasA after 16 hours of polymerization in buffer or AF.

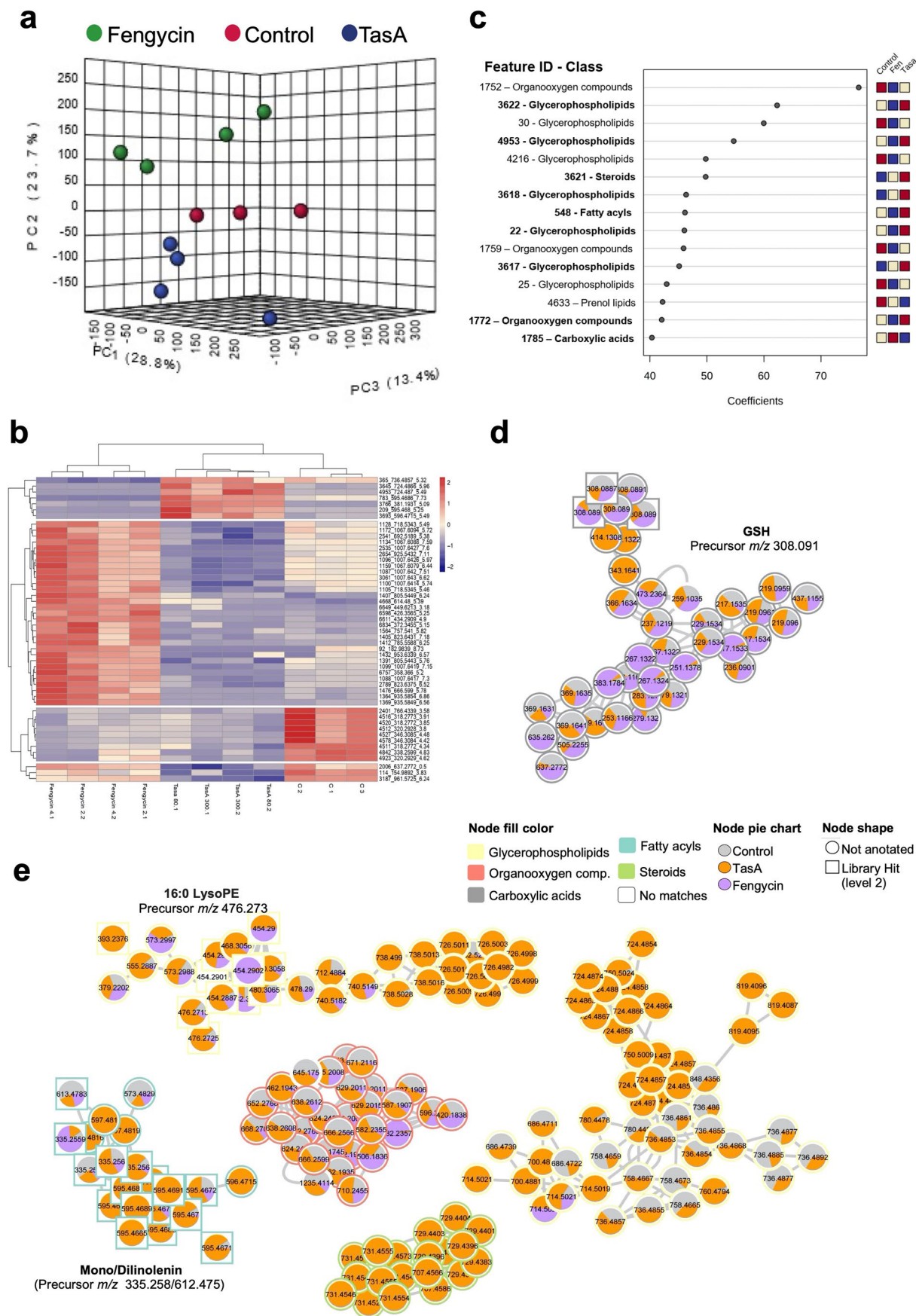

**Extended Data Fig. 6 | See next page for caption.**

**Extended Data Fig. 6 | Treatments of the seeds with TasA and fengycin influence the final metabolic patterns of emerged radicles. a** PCA 3D score plot of the metabolome of the radicles showing clustering of the samples based on seed treatment. The percentage of variation explained by each principal component is indicated on the axes. **b** Heatmap of hierarchical clustering results of the top 50 features impacted in the radicles grown from TasA- and fengycin-treated seeds. The color code inside the heatmap depicts the relative fold change of each metabolite between groups. **c** Top 20 features with high median weighted sum of absolute regression coefficient scores determined by PLS-DA and calculated using MetaboAnalyst. Feature IDs are accompanied by chemical class according to Classyfire classification. Features in bold are selected as features discriminating any growth-promoting treatment from the control treatment. **d, e** Molecular family of features discriminating fengycin-treated seeds (**D**) or TasA-treated seeds (**E**) from the control seeds according to PLS-DA. The chemical structures of annotated features and their average mass based on spectral matches to GNPS libraries are also represented for the corresponding molecular families. Pie charts indicate the peak abundance of each metabolite in the corresponding condition. Node shape indicates the level of identification according to Sumner et al., [78].

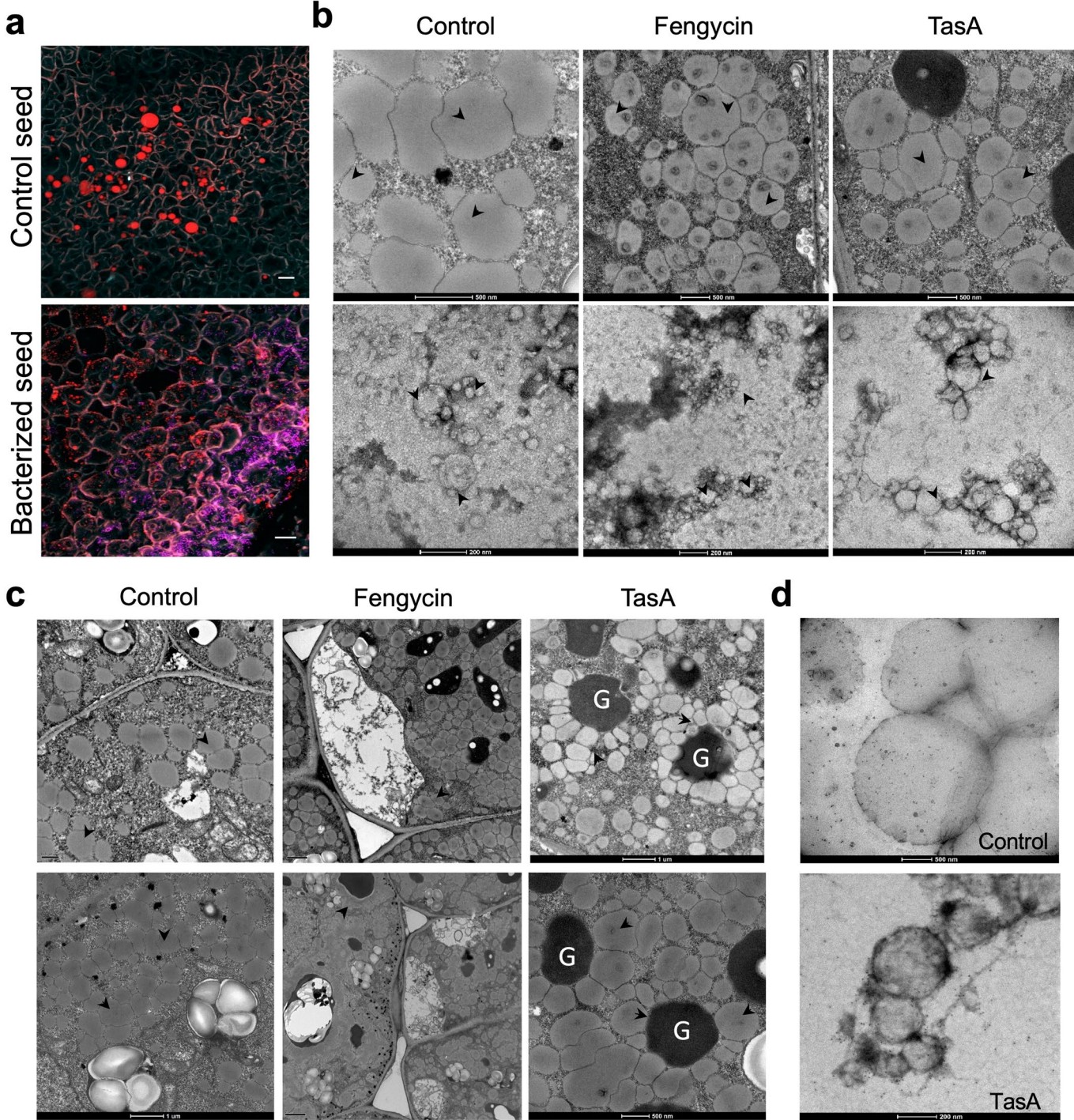

**Extended Data Fig. 7 | The size of OBs in the seeds is altered after seed treatment with *B. subtilis* cells or with purified TasA and fengycin.**
**a** Representative CLSM images of transversally cut seeds 16 hours after the treatment with *B. subtilis*. Oil bodies were stained with Nile red, and bacteria were stained with Hoechst. Scale bar: 20 μm. **b** Transmission electron microscopy images of thin sections of the seeds (top) or purified OB suspensions (bottom) 16 hours after the treatment with water (control), fengycin, or TasA. Arrows indicate oil bodies. **c** Representative transmission electron microscopy images of thin sections of the seeds 16 hours after the treatment with water (control), fengycin or TasA. G: glyoxysomes; arrows: oil bodies. **d** Transmission electron microscopy images of purified OB suspension 16 hours after the treatment with TasA. Samples were immunolabeled with a TasA antibody (1:150) and secondary anti-rabbit antibodies conjugated to 10 nm gold particles (1:50).

**a**

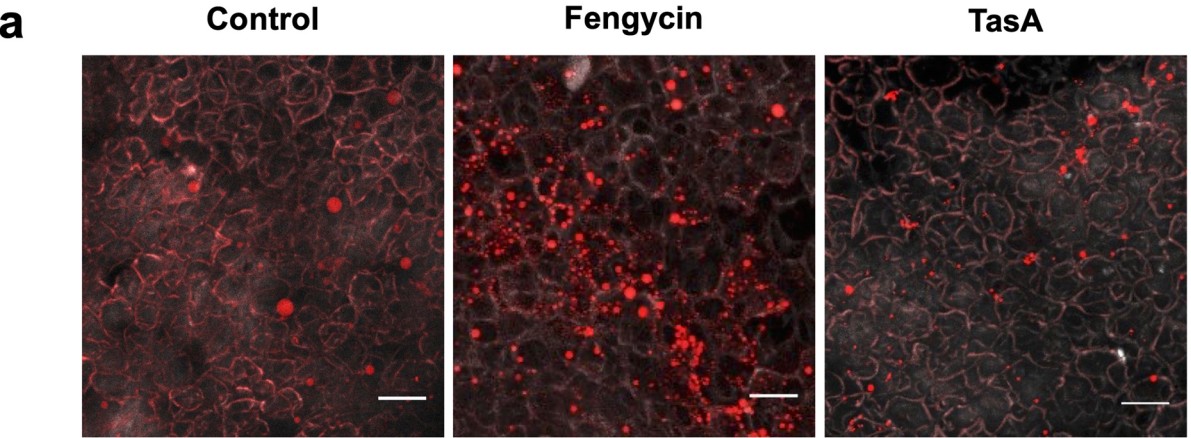

**b**

**Extended Data Fig. 8 | See next page for caption.**

**Extended Data Fig. 8 | TasA and fengycin induce changes in OBs morphology and glycerophospholipids dynamics in seeds. a** Representative CLSM images of transversally cut seeds 16 hours after the treatment with 10 μM fengycin or 3 μM TasA. Oil bodies were stained with Nile red (scale bar: 20 μm). **b** Chemical proportionality analysis of a glycerophospholipids molecular family in non-treated seeds (control) and seeds treated with fengycin or TasA from 0 h to 24 h after treatment. The size of the nodes is directly related with the abundance of the features at time 0 h. Arrows indicate chemical directionality of the modifications found and the color scale of the arrow represents the ChemProp score, the value obtained after the measurement of the peak area changes of connected nodes in a molecular network across a sequential data frame by comparing their proportions.

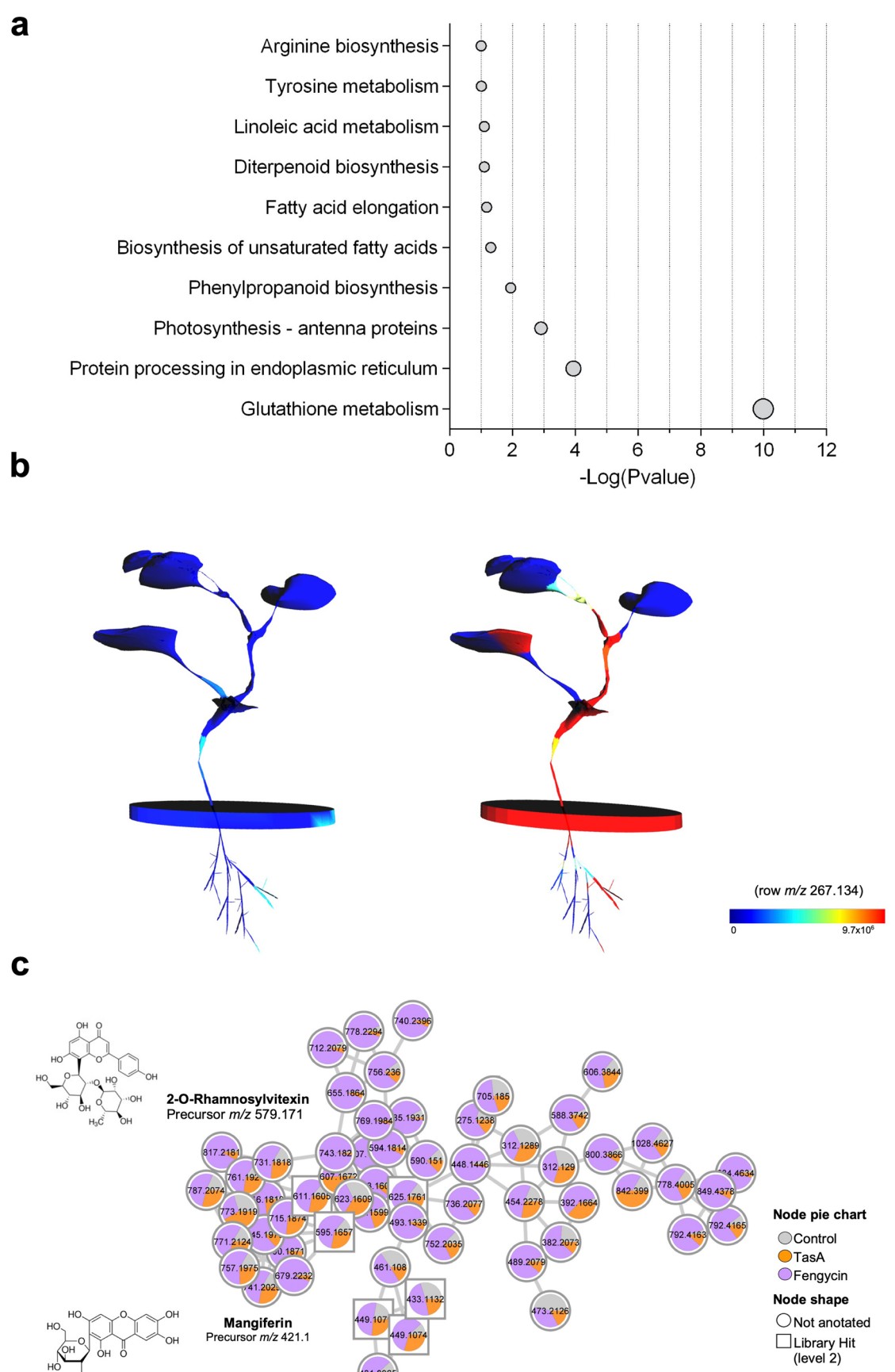

**a**

**b**

(row *m/z* 267.134)

0          9.7x10⁶

**c**

**2-O-Rhamnosylvitexin**
Precursor *m/z* 579.171

**Mangiferin**
Precursor *m/z* 421.1

Node pie chart
⚪ Control
🟠 TasA
🟣 Fengycin

Node shape
⚪ Not anotated
⬜ Library Hit (level 2)

**Extended Data Fig. 9 | See next page for caption.**

**Extended Data Fig. 9 | Treatment of the seeds with fengycin increase the expression and accumulation of metabolites related to glutathione metabolism and flavonoids. a**. KEGG pathway enrichment analysis of all DEGs identified by total transcriptomic analysis of the leaves of adult plants grown from control or fengycin-treated seeds 48 hours after inoculation with *B. cinerea*. **b** 3D plant models of adult plants grown from control (left) or *B. subtilis*-treated seeds (right) showing global distribution of the feature with *m/z* identical to discriminant feature 1785 of the GSH molecular family identified in the metabolome of the radicles grown from the seeds treated with purified TasA or fengycin. **c** Molecular family with features differentially abundant in the aerial regions of seedlings from fengycin-treated seeds vs. control seedlings 5 days after seed treatment, annotated as flavonoids according to Classyfire. The chemical structures of annotated features and their average mass based on spectral matches to GNPS libraries are also represented for the corresponding molecular families. Pie charts indicate the peak abundance of each metabolite in the corresponding condition. Node shape indicates the level of identification according to Sumner et al., [78].

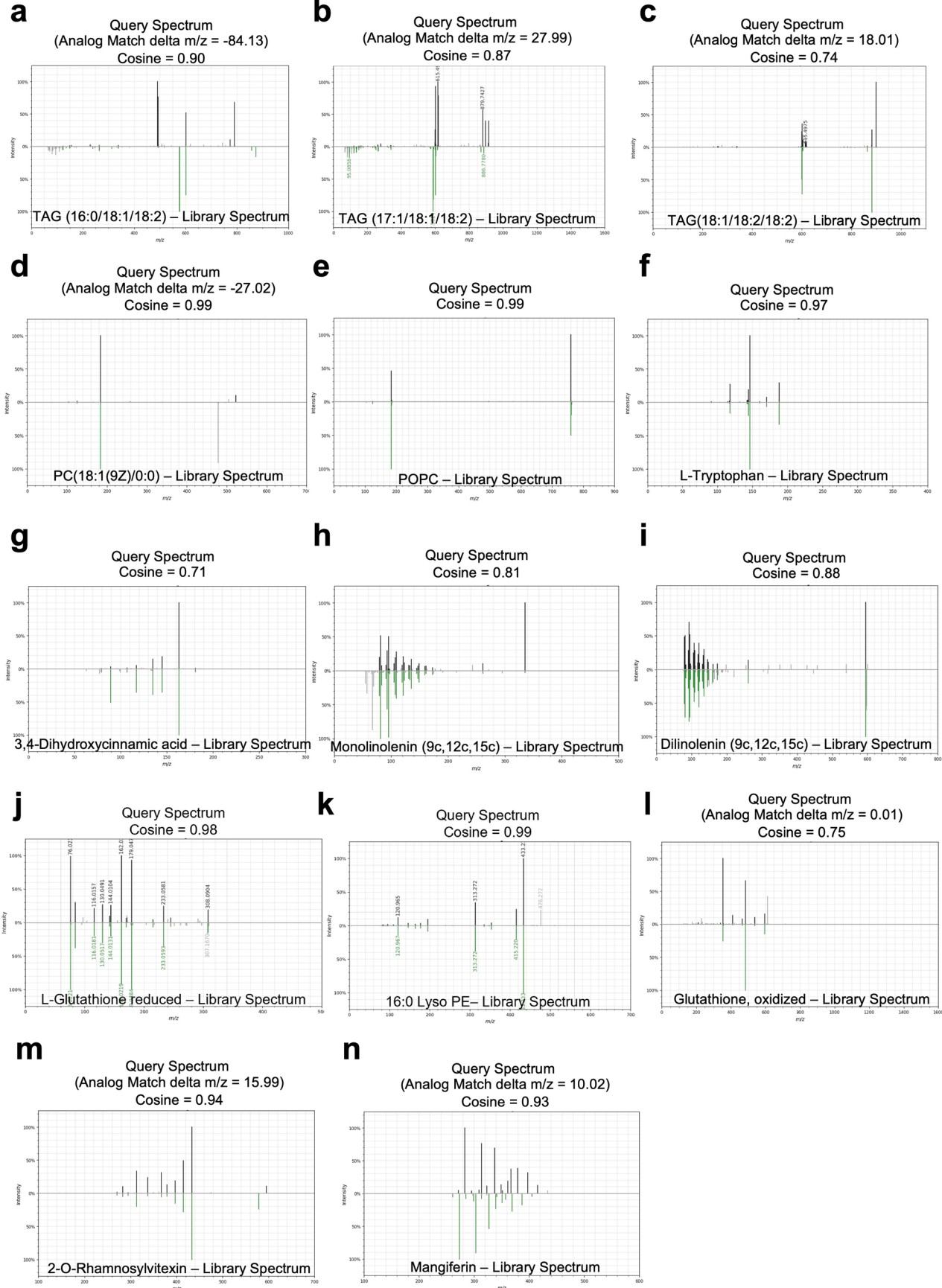

**Extended Data Fig. 10 | See next page for caption.**

**Extended Data Fig. 10 | Mirror plots comparing spectra from named features along the manuscript to standard spectra deposited in GNPS.** In the upper part of the plot (black lines) is represented the MS spectra of the candidate feature and in the lower part (green lines); the MS spectra of the standard compound. Mirror plots have been generated using https://metabolomics-usi.ucsd.edu/.

# Reporting Summary

## Statistics

For all statistical analyses, confirm that the following items are present in the figure legend, table legend, main text, or Methods section.

| n/a | Confirmed | |
|---|---|---|
| ☐ | ☒ | The exact sample size (*n*) for each experimental group/condition, given as a discrete number and unit of measurement |
| ☐ | ☒ | A statement on whether measurements were taken from distinct samples or whether the same sample was measured repeatedly |
| ☐ | ☒ | The statistical test(s) used AND whether they are one- or two-sided *Only common tests should be described solely by name; describe more complex techniques in the Methods section.* |
| ☒ | ☐ | A description of all covariates tested |
| ☒ | ☐ | A description of any assumptions or corrections, such as tests of normality and adjustment for multiple comparisons |
| ☐ | ☒ | A full description of the statistical parameters including central tendency (e.g. means) or other basic estimates (e.g. regression coefficient) AND variation (e.g. standard deviation) or associated estimates of uncertainty (e.g. confidence intervals) |
| ☐ | ☒ | For null hypothesis testing, the test statistic (e.g. *F*, *t*, *r*) with confidence intervals, effect sizes, degrees of freedom and *P* value noted *Give P values as exact values whenever suitable.* |
| ☒ | ☐ | For Bayesian analysis, information on the choice of priors and Markov chain Monte Carlo settings |
| ☒ | ☐ | For hierarchical and complex designs, identification of the appropriate level for tests and full reporting of outcomes |
| ☒ | ☐ | Estimates of effect sizes (e.g. Cohen's *d*, Pearson's *r*), indicating how they were calculated |

*Our web collection on statistics for biologists contains articles on many of the points above.*

## Software and code

Policy information about availability of computer code

| | |
|---|---|
| Data collection | RNA-seq data were collected using NextSeq System Suite v2.2.0. Confocal microscopy images were taken using Leica Application SuiteAdvance Fluorescence v2.7.3.9723. Electron microscopy images were taken using TIA FEI Imaging Software v4.14. Metabolomics data were obtained with the commercial software of Thermo Scientific for UPLC systems. HPLC-ESI-MS/MS data corresponding to the identification of proteins in gel cut bands was acquired using ProteinScape 3 software from Bruker coupled to Mascot v3.1 (Matrix Science). |
| Data analysis | For statistics and representation of all experimental data, analysis were done using GraphPad Prism v.6.01. <br><br>Raw reads were preprocessed by NextSeq System Suite v2.2.0. using specific NGS technology configuration parameters. Clean reads of the BAM files were aligned and annotated using the Cucumis melo genome (v4.0) as the reference by Bowtie2; these data were then sorted and indexed using SAMtools v1.48411074. Uniquely localized reads were used to calculate the read number values for each gene by Sam2counts (https://github.com/vsbuffalo/sam2counts). Differentially expressed genes (DEGs) in the treatment samples were analyzed by DEgenes Hunter, which provides a combined p value calculated based on the Fisher method using nominal p values provided by edgeRand DEseq2. This combined p value was adjusted by the Benjamini-Hochberg (BH) procedure (false discovery rate approach) and used to rank all obtained differentially expressed genes. A heatmap and DEG clustering were generated using ComplexHeatmap in R Studio and Kobas 2.0. DEGs annotated using the Cucumis melo genome were processed to identify the Gene Ontology functional categories using sma3s and TopGo software. KEGG pathways and enrichment were estimated using Bioconductor packages (Bioconductor.org) GGplot2, ClusterProfiler, DOSE, and EnrichPlot in R Studio. <br><br>For LC-MS/MS data analysis, raw spectra were converted to .mzXML files using MSconvert (ProteoWizard). MS1 and MS/MS feature extraction was performed with Mzmine2.30. For molecular networking and spectrum library matching the .mgf file was uploaded to GNPS |

(gnps.ucsd.edu). Molecular networks were visualized with Cytoscape v. 3.4.

For CLSM data analysis, image processing was performed using Leica LAS AF (LCS Lite, Leica Microsystems) and ImageJ 1.51s.

For manuscripts utilizing custom algorithms or software that are central to the research but not yet described in published literature, software must be made available to editors and reviewers. We strongly encourage code deposition in a community repository (e.g. GitHub). See the Nature Portfolio guidelines for submitting code & software for further information.

## Data

Policy information about availability of data

All manuscripts must include a data availability statement. This statement should provide the following information, where applicable:
- Accession codes, unique identifiers, or web links for publicly available datasets
- A description of any restrictions on data availability
- For clinical datasets or third party data, please ensure that the statement adheres to our policy

The RNA-seq data were deposited in the GEO database under the reference GSE175611.

Metabolomics data are deposited at https://massive.ucsd.edu/ with the identifiers MSV000084674 (data from radicles from seeds bacterized with Bacillus subtilis subsp subtilis NCBI 3610 and mutant strains; https://massive.ucsd.edu/ProteoSAFe/dataset.jsp?task=b2e3636335d24c2da72634b7c6c8b63f), MSV000084278 (data from radicles from the seeds treated with fengycin and TasA; https://massive.ucsd.edu/ProteoSAFe/dataset.jsp?task=fc05e1b986cb4073b56d9f9a044f0a2f), MSV000086360 (data from adult plants grown from control seeds or seeds bacterized with Bacillus subtilis subsp subtilis NCBI 3610; https://massive.ucsd.edu/ProteoSAFe/dataset.jsp?task=1090c82889a04e4ba9be1058a8a5d6db) and MSV000088139 (data from seeds and seedlings treated with fengycin, TasA or bacteria; https://massive.ucsd.edu/ProteoSAFe/dataset.jsp?task=72bb762b7b8d45da8d3c5c9005a425d8).

Cucurbit genome databases used for mass spectrometry analysis of protein bands were UniProtKB/TrEMBL TrEMBL Cucumis melo, v 2017.10.25.

# Field-specific reporting

Please select the one below that is the best fit for your research. If you are not sure, read the appropriate sections before making your selection.

☒ Life sciences        ☐ Behavioural & social sciences        ☐ Ecological, evolutionary & environmental sciences

For a reference copy of the document with all sections, see nature.com/documents/nr-reporting-summary-flat.pdf

# Life sciences study design

All studies must disclose on these points even when the disclosure is negative.

| | |
|---|---|
| Sample size | Sample-size calculation was not initially performed but a sufficient number of replicates were taken, prior experience with these types of data, and analysis of the current data, which indicates the reproducibility of the data. The number of replicates is sufficient in order to get statistically confident results, as demonstrated in the manuscript. In all the experiments performed, standard deviation and standard error were calculated and the corresponding statistical tests were performed when needed. |
| Data exclusions | No data were excluded from the study. |
| Replication | At least three replicates were taken in all the experiments performed. In all the cases the experiments were successfully replicated |
| Randomization | Randomization is not relevant in this case as our study is mainly focused in a plant-bacteria interaction. |
| Blinding | Blinding is not relevant in this case as our study is mainly focused in a plant-bacteria interaction. |

# Reporting for specific materials, systems and methods

We require information from authors about some types of materials, experimental systems and methods used in many studies. Here, indicate whether each material, system or method listed is relevant to your study. If you are not sure if a list item applies to your research, read the appropriate section before selecting a response.

## Materials & experimental systems

| n/a | Involved in the study |
|-----|----------------------|
| ☐ | ☒ Antibodies |
| ☒ | ☐ Eukaryotic cell lines |
| ☒ | ☐ Palaeontology and archaeology |
| ☒ | ☐ Animals and other organisms |
| ☒ | ☐ Human research participants |
| ☒ | ☐ Clinical data |
| ☒ | ☐ Dual use research of concern |

## Methods

| n/a | Involved in the study |
|-----|----------------------|
| ☒ | ☐ ChIP-seq |
| ☒ | ☐ Flow cytometry |
| ☒ | ☐ MRI-based neuroimaging |

## Antibodies

| | |
|---|---|
| Antibodies used | The anti-TasA antibody used for immunodetection studies in this work was kindly provided by professor Adam Driks (Stover and Driks, 1999) and it consists in a polyclonal antiserum obtained from blood 23 days after injecting the protein into rabbbit. The secondary antibody used in western blot was commercially available at Bio-Rad (Goat Anti-Rabbit IgG (H + L)-HRP Conjugate cat. no. 1706515). The immunogold-conjugated secondary antibody used was coomercially available at BBI solutions (EM Grade 20nm Goat anti-Rabbit Conjugate cat. no. EM.GAR20/1). |
| Validation | The anti-TasA primary antibody was validated elsewhere in the literature (Stover and Driks, 1999). The Goat Anti-Rabbit IgG (H + L)-HRP Conjugate (cat. no. 1706515) secondary antibody from BioRad was validated by the supplier as follows: "Specific for rabbit IgG, heavy and light chain. The cross-reactivities of anti-rabbit IgG antibody are tested in an ELISA. Minimum cross-reactivity to human IgG. 1:2,000-1:5,000 dilution can be used (Coligan, J., 1997)". |

