## [Peer Review File · Nature Microbiology]

Peer Review Information

Journal: Nature Microbiology

Manuscript Title: Bacillus subtilis biofilm matrix components target seed oil bodies to promote growth and anti-fungal resistance in melon

Corresponding author name(s): Diego Romero

Reviewer Comments & Decisions:

Decision Letter, initial version:

Dear Professor Romero,

Thank you for your patience while your manuscript "The crosstalk between Bacillus extracellular matrix and seed lipid storages stimulates overgrowth and immunization of plants" was under peer review at Nature Microbiology. Please first accept my sincere apologies for the delay in getting back to you with a decision, but unfortunately some of the referees were quite delayed in returning their comments to us. Nonetheless, the paper has now been seen by 3 reviewers, whose expertise and comments you will find at the end of this email. In the light of their advice, I'm sorry to say that we have decided that at this stage we cannot offer to publish your manuscript in Nature Microbiology.

From the reports, you will see that while the referees are generally enthusiastic and find your work of potential interest, they also raise substantial concerns about the strength of the novel conclusions that can be drawn at this stage. In particular, each reviewer goes on to point out different areas where additional work is needed in order to be able to substantiate the main claims. For example, they point out the need to rule out alternative mechanisms (like phytohormones); to show that the effects are happening in the seeds and not via colonization of the adult plant; to clarify the methods used to identify LysoPEs and GSH; and to further elucidate the molecular mechanisms responsible for the observed phenotypes (such as those underpinning the impact of TasA/fengycin and changes to OBs; and how this may lead to changes in the adult plant). Given the length of time and substantial commitment that it would likely take to address these concerns thoroughly, we felt that these criticisms are currently sufficiently important as to preclude further consideration of your work in Nature Microbiology. However, as mentioned above, we certainly acknowledge the topic interest expressed by the referees, so would like to note that if you do feel that you would be able to include additional work to address these concerns, we would be willing to consider an appeal at a later stage (although please note that we would reassess novelty with respect to existing literature at the time of appeal and would be unlikely to trouble the referee again unless we felt that their concerns had been satisfied). In the case of a successful appeal and eventual publication, the received date would be that of the revised paper.

I am sorry that we cannot be more positive on this occasion, but hope that you find the referees' comments helpful when revising your paper.

With best regards,

{redacted}

2Reviewer Expertise:

Referee #1: Rhizosphere/plant interactions; PGPR cellular and molecular biology; Bacillus

Referee #2: Bacillus cellular and molecular biology; biofilms; Bacillus/plant interactions

Referee #3: Plant-microbe interactions; multi-omics

Reviewers Comments:

Reviewer #1 (Remarks to the Author):

Manuscript # NMICROBIOL-21092279

Title: The crosstalk between Bacillus extracellular matrix and seed lipid storages stimulates overgrowth and immunization of plants

PGPR strains can promoted seed germination and plant growth through various known mechanisms, such as production of phytohormones and many other signals to stimulate plant root development, solubilizing unavailable nutrients et al. they can also induce plant resistance against pathogen invasion. In this study, an interesting exploration of how the Bacillus subtilis NCIB3610 stimulate the seed germination and radicle emergency was conducted, the authors suggested that biofilm matrix components lipopeptide fengycin and amyloid protein TasA mediated chemical dialogue resulted the promotion of radicle growth after seed germination, overgrowth of adult plants and protection against the aerial necrotrophic fungus Botrytis cinerea. Both molecules targeted the lipid storage oil bodies of the seed endosperm, reprogrammed specific metabolites.

In general, this exploration is interesting, but some conclusions are overstated, which are kind of speculative without support of very strong evidences. Also, other possible mechanisms contributed to the overgrowth and immunization of plants by this strain are even not considered.

Major comments:

1) PGPR strain produced phytohormones are well-known for plant development, including seed germination, seedling development et al., even the plant genes for ABA and GA synthesis were not changed, but PGPR strains are usually produce phytohormones themselves to stimulate seed germination or plant development. I don't how to exclude the bacterial produced phytohormones' activity in figure 1, 2, 3 and related supplementary figures?

2) Metabolome is an important strategy in this study, the top 50 metabolites are usually selected for further testing, but for regulation of plant development, it is usually not linear positive with the metabolite concentration, the phytohormones are usually showed extremely low level, maybe even under the detection limitation of metabolome, but their stimulation activity for plant development is very strong.

3) The conclusion that "B. subtilis triggered genetic and physiological responses in the seeds that determined subsequent metabolic and developmental status of adult plants" is arbitrary, since the results showed B. subtilis can colonize and persist in the seed and radicle, it is expected that they can still persist in the plant seedling and adult plant, which was demonstrated in many studies, so it is possible that B. subtilis can directly interact with the adult plants to determine their metabolic and

2developmental status, it is not necessary to determine these post-seed development in the seeds stage.

4) For the same reason, treatment of seeds with fengycin elicits plant defense against *Botrytis cinerea* may also be the activity of residual fengycin in the adult plant stage.

5) When testing the interaction of TasA and fengycin with the seed oil bodies, *B. subtilis* produced other SMs should also be included, as proposed, fengycin can induce the formation of the pores and subsequent changes in membrane permeability, *B. subtilis* produced surfactins also showed this function in previous reports of this strain. Currently, the functional mechanism of fengycin is mostly speculated.

6) line 173-176, these conclusions are overstated, supporting evidences are not enough.

7) line 264, Fig. 1F should be Fig. 3F.

Reviewer #2 (Remarks to the Author):

This manuscript describes the effect of *B. subtilis* on radicles emergence and overall plant growth following seed inoculation. The main founding that fengycin and the TasA protein have specific effect on the seed and emerging seedling is examined by many techniques, and extensive metabolomic analysis. These observations are undoubtedly novels and deeply explored. However, the manuscript is very dense, and figures/results appear somewhat disconnected from one another. Certain conclusions are also not well supported.

The authors did a great job at showing effect of TasA on OBs and the subsequent metabolomic changes observed, but there is little evidence that the fengycin's effect on OBs leads to GSH production and persistence of growth promotion and immunization in adult; the explanation stated in line 326 to 329 is only hypothetical. Since the effect of TasA on OBs has no long-term effect, it is unclear why/how effect of fengycin on OBs would lead to effect in adult plant. It appears equally likely that fengycin has other effect(s) in addition to OBs, which could promote GSH production and persist in adult plants. The table S6 hints that the effect of fengycin might be through OBs, but it is a very indirect proof, the table does not contain data information, and certain Cucurbitaceae are not promoted by fengycin. Consequently, the title and the conclusions stating that crosstalk with lipid storages stimulates overgrowth in adult plant and immunization is overstated.

There is a disconnection between the analysis of metabolomic profile of the various mutants (2bcd) and of the metabolomic profiles resulting from fengycin or TasA application (S8). To demonstrate that fengycin and TasA are the molecule having an impact when seeds are treated with *B. subtilis*, metabolomic profiles should be analyzed together, since there should be at least some redundancy between the Δ tasA mutant and fengycin application, for example.

Similarly, a parallel between Fig 1b (transcriptomic of seeds) and Fig 4 (effect of fengycin and TasA on OBs) would help validate with a transcriptomic signature that OBs are affected by *B. subtilis* colonization of seeds.

Why the Δ pks and Δ ppls strains failing to promote radicle growth if they are still producing TasA, and

3for Δ pss, fengycin?

Fig 2abc. Why is the molecular signature of Δ tasA mutant more similar to non-growth promoting mutants than 3610? Shouldn't it be closer to the 3610 signature, since it overproduces fengycin and stimulates radicle growth?

Fig1de and S2 are great, but they do not bring important information to this already dense manuscript and could be part of an independent article.

In Fig 1g, the word "colonization" would be better than "persistence", since for persistence the % should represent difference between early CFU (e.g. 4h or 1 day) vs 5 day CFU, but colonization would refer to a comparison with WT CFU. Adding Figure S4b (early colonization) to the main figures would also favor a more complete understanding of the short term / long term colonization capacity of mutants.

Fig 4a. Putting the images quantification (S10d) as a main figure would help a lot the interpretation of this panel.

Fig S1B. The color scale needs to be specified; is it log2?

Fig S3A. Resolution of the figure is terrible, and I cannot see purple and gray highlights

Fig S3B. Table is partially cut off by a blank region

Line 126-128. Usage of "instead" makes the sentence unclear, please revisit.

Names of the mutants need to be put in italic throughout the manuscript

Line 146. Specified that it is in the seed extract

Line 283. This sentence refers to bacterial-treated seeds right? It needs to be specified.

Line 310-313. I do not understand how the authors make a direct link between TasA-oleosin interaction and lysophospholipids accumulation. Is there any literature that can back this?

Reviewer #3 (Remarks to the Author):

Overall, Berlanga-Clavero et al. present a compelling study that utilizes a truly staggering number of experiments and techniques. They find that *B. subtilis*, and specifically, ECM components produced by *B. subtilis*, have play role in enhancing the growth of melon plants. Moreover, they show that this apparently relies on interaction of fengicins and TasA protein with OBs in the seeds post germination. I see this as a landmark study since it covers so much ground toward explaining how microbes can enhance seedling growth at the molecular level. As such, I am broadly enthusiastic about this work.

4That said, I have several concerns listed below.

Major concerns:

I have strong reservations about the quantitative nature of the mass spectrometry data. As written, it is unclear to me whether or not the metabolomic data was acquired in a way that enables quantitative comparisons. I think what is missing is an explanation of what steps were taken to ensure that a quantitative, or semi-quantitative analysis, is possible here. For example, was the same amount of biomass extracted across all samples? Or, if not, were the results normalized to the biomass amount? How were spike-in controls used to normalize the data? How many replicates were there and how were they handled at the analysis stage? Were extraction controls used?

P10 L234: What is the basis for the identification of lysophosphatidylethanolamines (LysoPEs) glutathione (GSH)? The figure shows the m/z (presumably the H^+ adduct?) of each of these molecules (481.6 and 307.32, respectively), however, as far as I can tell, those m/z s are not present in the networks shown. Comparisons MS/MS fragmentation patterns and retention time of the detected ions with authentic standards should be shown in the SI. The level of identification according to Sumner et al 2007 should be reported. I should note that while I believe that the further experiments validate the LysoPE and GSH findings, right now the requirements for positive IDs of these compounds in the metabolomics data sets are not being met.

Lines 285-286 and Figure 4A: These oil body micrographs are difficult to interpret. I see more bodies in the Fengycin and TasA treated seeds, and the bodies are also smaller. The control has comparatively few OBs. Would disaggregation of the few OBs in the control lead to the distributions seen in the Fengycin and TasA samples? I don't know, but it seems difficult to make that conclusion based on these images. When things look ambiguous like this, I think the best course of action is to look at many samples and quantify the effect. I think some quantification here would go a long way toward making this more convincing.

Minor Concerns:

Use of the term 'overgrowth': Overgrowth seems like it might imply a phenotype that has negative overall effect on the plants. Here the plants (radicles and above-ground parts) just seem to have enhanced growth, with no negative cost. For that reason, I would consider 'enhanced growth' rather than 'overgrowth'.

P10 L195: "an increase in reduced abundance of the prenol lipid and stilbene": this sentence is unclear. Is the abundance increased or reduced?

Figure 1E: I found this figure a bit confusing. First, the color legend is so small it is almost impossible to read. Second, what exactly are we looking at here? The plant shape is exactly identical between the control and treated conditions. This leads me to conclude that the sample data was mapped onto a generic plant shape. If this is true, then I am questioning the validity of this representation. For example the treated one is red on two of the four leaves. Why these two and not the others? This is a cool way of looking at the data, but I am not convinced that this is the most objective way of

5displaying it. These distributions correspond to specific MS features. What are these features, and why are they important?

Figure 2B: I see the colors red and blue, and I see a tiny legend that shows the scale goes from roughly 2 to -2. The problem is that I don't know what this scale means. Is this fold change? Log fold change? This appears to be a quantitative analysis, but how this should be interpreted is unclear. Does red to white represent a statistically significant change? Beyond this, we have little idea what these feature represent in terms of their chemical classes.

Figure 3B: The nodes are colored as orange, purple, and grey pie charts. What determines each sample's slice of the pie? Is it the number of times that feature was detected in each sample? The peak area in each sample? How were the replicates treated for this experiment? Frustrating since it seems to mean something but it is not explained in the figure legend.

Author Rebuttal to Initial comments

Reviewer #1:

PGPR strains can promoted seed germination and plant growth through various known mechanisms, such as production of phytohormones and many other signals to stimulate plant root development, solubilizing unavailable nutrients et al. they can also induce plant resistance against pathogen invasion. In this study, an interesting exploration of how the Bacillus subtilis NCIB3610 stimulate the seed germination and radicle emergency was conducted, the authors suggested that biofilm matrix components lipopeptide fengycin and amyloid protein TasA mediated chemical dialogue resulted the promotion of radicle growth after seed germination, overgrowth of adult plants and protection against the aerial necrotrophic fungus Botrytis cinerea. Both molecules targeted the lipid storage oil bodies of the seed endosperm, reprogrammed specific metabolites.

In general, this exploration is interesting, but some conclusions are overstated, which are kind of speculative without support of very strong evidences. Also, other possible mechanisms contributed to the overgrowth and immunization of plants by this strain are even not considered.

R- We do really appreciate the reviewer's introductory comment. It certainly was not our intention with this work to exclude all the knowledge accumulated over several studies on the role of bacterial molecules in the beneficial interaction of bacteria with plants. Seed germination and radicle growth are really complex biological processes tightly regulated by interconnected genetic and metabolic pathways which ensure further vegetative growth of plants in a supporting environment. Thus, we have focused on aspects of seed germination poorly explored or experimentally approached with a different angle. In

6our study we bring new knowledge on the role of the bacterial ECM beyond the well-studied structural functionality, in the establishment of a mutualistic interaction with a plant. From the plant side, we offer complementary metabolic knowledge to the well-studied role of plant hormones during the first stages of plant development from seeds to vegetative growth. We have been able to define metabolic signatures that may decipher a process of plant growth stimulation and even immunization, and it is our belief that these findings will be largely appreciated by the scientific community dedicated to the study of this fascinating stage of plant growth, alone or in the interaction with microbes.

Major comments:

1) PGPR strain produced phytohormones are well-known for plant development, including seed germination, seedling development et al., even the plant genes for ABA and GA synthesis were not changed, but PGPR strains are usually produce phytohormones themselves to stimulate seed germination or plant development. I don't how to exclude the bacterial produced phytohormones' activity in figure 1, 2, 3 and related supplementary figures?

R- As mentioned by the reviewer, phytohormones are important molecules in plant development. Although the role of these molecules produced by bacteria has been widely described, we have not found much information regarding key biosynthetic genes in *Bacillus subtilis* NCIB 3610. However, In order to analyze the possible presence of phytohormones in our metabolomic datasets, we have manually searched m/z masses with no satisfactory results. In addition, in an attempt to determine the intrinsic ability of *B. subtilis* to produce these molecules, we have analyzed other metabolomes of *B. subtilis* and *Pseudomonas* (See Molina-Santiago et al., 2019), another PGPR strain known to produce phytohormones, growing on LB media using the same methodology described in our manuscript. In this additional analysis we have found features that match with jasmonic acid in both strains at different time points indicating that *B. subtilis* is able to produce, at least, jasmonic acid related molecules. The fact that we are not able to detect these molecules in our analysis in planta, suggests that if this molecule is produced, it must be done at a really low level.

In addition, I would like to emphasize that the message of our manuscript is related to the contribution of *Bacillus* ECM in maintaining a mutualistic interaction, that we do know is multifactorial and dependent on diverse molecules. Nevertheless, although we don't exclude a role of phytohormones produced by *Bacillus* in the stimulation of seed germination or plant development, our study adds fengycin and TasA to the pool of bacterial molecules that contribute to the health of the plants at the very initial stages of seeds. Based on our data, it appears that at least these two ECM components complement the promotion of seed germination associated with volatiles or phytohormones, by accelerating the growth of the emerging radicles.

2) Metabolome is an important strategy in this study, the top 50 metabolites are usually selected for further testing, but for regulation of plant development, it is usually not linear positive with the metabolite concentration, the phytohormones are usually showed extremely low level, maybe even under the detection limitation of metabolome, but their stimulation activity for plant development is very strong.

R- Although we have selected generally the top 50 changing metabolites for representation, we have analyzed the whole metabolome and, if relevant, we have highlighted non-top 50 metabolites. As mentioned above, phytohormones were not detected in the whole metabolomes, and together with the fact that the focus of this study was totally different, we did not mention these compounds as possible modulators of the promoting effect. Nevertheless, keeping this interesting comment in mind, we have re-written the text to reference the effect of phytohormones (see Lines 427-431).

In addition, in the original Fig S3 we had incorporated heatmaps of the top 50 metabolites with the highest relative fold changes to highlight general changes between treatments. Considering this comment and some others of reviewer 2 we have changed it to volcano plots with all the metabolites impacted to better transmit this general idea (See new Fig S3).

3) The conclusion that "B. subtilis triggered genetic and physiological responses in the seeds that determined subsequent metabolic and developmental status of adult plants" is arbitrary, since the results showed B. subtilis can colonize and persist in the seed and radicle, it is expected that they can still persist in the plant seedling and adult plant, which was demonstrated in many studies, so it is possible that B. subtilis can directly interact with the adult plants to determine their metabolic and developmental status, it is not necessary to determine these post-seed development in the seeds stage.

R- In our study we have observed that *Bacillus* population completely sporulated after 5 days of after seed soaking in cell suspension, and in the emerging radicle the bacterial population was also sporulated. We did not exclude the possibility that these spores may enter a round of germination, but the number of vegetative cells might be considerably reduced. In any case, we have done new experiments to study the presence of *B. subtilis* in leaves and stems of plants from bacterized and non-bacterized seeds 25 days after treatment. CFU counts of these sections have shown non-detectable levels of *B. subtilis* cells, which support our results and conclusions. In addition to this finding, it is important to notice that we have been able to reproduce the promotion of radicle or plant growth with the addition of purified fengycin or TasA to the seeds.

4) For the same reason, treatment of seeds with fengycin elicits plant defense against *Botrytis cinerea* may also be the activity of residual fengycin in the adult plant stage.

R- In order to confirm the absence of fengycin (added or produced by *B. subtilis*) in adult plant stages, we searched masses corresponding to fengycin in our metabolomic datasets from radicles of treated seeds (with bacteria or pure fengycin) 5 days after treatment. We also analyzed leaf, stem and root sections of adult plants from bacterized seeds. However, we did not find any feature annotated as fengycin/piplastatin in any of the metabolomes analyzed.

In a new metabolomic analysis (MSV000088139), which has also been included in the new version of the manuscript, we analyzed seeds from 0 days after treatment. Although we did not find any annotated feature corresponding to fengycin with an abundance higher to the defined threshold (2E2), the analysis of raw data in MzMine2.53 eliminating the threshold for noise levels showed a peak with a m/z and retention time concordant with the fengycin. This peak, detected at a retention time of 4.8 min, was not detected in non-treated seeds, nor aerial regions or radicles five days after treatment of the seeds with fengycin (Figure R1a,b). When comparing these samples with other samples where these compounds are identified as a control (<https://www.nature.com/articles/s41467-019-09944-x>, MSV000082402) using the GNPS dashboard (<https://www.nature.com/articles/s41592-021-01339-5>) we can appreciate that the abundance of the putative fengycin is extremely low (Figure R1c). The slight differences observed in the retention time between samples from MSV000088139 and the control sample used are probably due to utilization of two different LC-MS/MS equipment and columns (The analysis can be checked using the following link: [https://gnps-
lcms.ucsd.edu/?xic_mz=1463.67&xic_formula=&xic_peptide=&xic_tolerance=0.5&xic_ppm_tolerance=10&xic_tolerance_unit=ppm&xic_rt_window=&xic_norm=False&xic_file_grouping=MZ&xic_integration_type=AUC&show_ms2_markers=True&ms2marker_color=blue&ms2marker_size=5&ms2_identifier=Non
e&show_lcms_2nd_map=False&map_plot_zoom=%7B%7D&polarity_filtering=None&polarity_filtering2
=None&tic_option=TIC&overlay_usi=None&overlay_mz=row+m%2Fz&overlay_rt=row+retention+time&
overlay_color=&overlay_size=&overlay_hover=&overlay_filter_column=&overlay_filter_value=&feature
finding_type=Off&feature_finding_ppm=10&feature_finding_noise=10000&feature_finding_min_peak
rt=0.05&feature_finding_max_peak_rt=1.5&feature_finding_rt_tolerance=0.3&synchronization_session
id=63d24ab3138a4ae8b18ac4acc1939cef&chromatogram_options=%5B%5D&comment=&map_plot_c
olor_scale=Hot_r&map_plot_quantization_level=Medium&plot_theme=plotly_white#%7B%22usi%22%
3A%20%22mzspec%3AMSV000088139%3Apeak/20210823_MV01.mzXML%5Cnmzspec%3AMSV000088
139%3Apeak/20210823_MV19.mzXML%5Cnmzspec%3AMSV000088139%3Apeak/20210823_MV49.mzX
ML%5Cnmzspec%3AMSV000088139%3Apeak/20210823_MV67.mzXML%5Cnmzspec%3AMSV00008240
2%3Apeak/mxzml/3610_LB_24h_02x.mzXML%5Cn%22%2C%20%22usi_select%22%3A%20%22mzspec%
3AMSV000088139%3Apeak/20210823_MV01.mzXML%22%2C%20%22usi2%22%3A%20%22%22%7D](https://gnps-
lcms.ucsd.edu/?xic_mz=1463.67&xic_formula=&xic_peptide=&xic_tolerance=0.5&xic_ppm_tolerance=10&xic_tolerance_unit=ppm&xic_rt_window=&xic_norm=False&xic_file_grouping=MZ&xic_integration_type=AUC&show_ms2_markers=True&ms2marker_color=blue&ms2marker_size=5&ms2_identifier=Non
e&show_lcms_2nd_map=False&map_plot_zoom=%7B%7D&polarity_filtering=None&polarity_filtering2
=None&tic_option=TIC&overlay_usi=None&overlay_mz=row+m%2Fz&overlay_rt=row+retention+time&
overlay_color=&overlay_size=&overlay_hover=&overlay_filter_column=&overlay_filter_value=&feature
finding_type=Off&feature_finding_ppm=10&feature_finding_noise=10000&feature_finding_min_peak
rt=0.05&feature_finding_max_peak_rt=1.5&feature_finding_rt_tolerance=0.3&synchronization_session
id=63d24ab3138a4ae8b18ac4acc1939cef&chromatogram_options=%5B%5D&comment=&map_plot_c
olor_scale=Hot_r&map_plot_quantization_level=Medium&plot_theme=plotly_white#%7B%22usi%22%
3A%20%22mzspec%3AMSV000088139%3Apeak/20210823_MV01.mzXML%5Cnmzspec%3AMSV000088
139%3Apeak/20210823_MV19.mzXML%5Cnmzspec%3AMSV000088139%3Apeak/20210823_MV49.mzX
ML%5Cnmzspec%3AMSV000088139%3Apeak/20210823_MV67.mzXML%5Cnmzspec%3AMSV00008240
2%3Apeak/mxzml/3610_LB_24h_02x.mzXML%5Cn%22%2C%20%22usi_select%22%3A%20%22mzspec%
3AMSV000088139%3Apeak/20210823_MV01.mzXML%22%2C%20%22usi2%22%3A%20%22%22%7D)).

9Based on these findings, we conclude that no residual fengycin is able to persist until later stages of seed germination.Figure R1

a

b

c

XIC Plot - Grouped Per M/Z

Figure R1. Extracted ion chromatograms (XIC) plots from raw data in MzMine2.53 of metabolites in the range 1460-1470 m/z in **a**) samples from micropylar regions of control seeds (blue lines) and fengycin-treated seeds (red lines) immediately after treatment and **b**) samples from aerial regions of seedlings from control seeds (blue lines) and fengycin-treated seeds (red lines) five days after treatment. The highlighted region corresponds to the range of retention times where fengycins/plipastatins are detected according to other metabolomics datasets. **c** XIC plot of metabolites with a m/z of 1463.67 with tolerance 10 ppm in samples from micropylar regions of control seeds (blue line), fengycin-treated seeds (red line) immediately after treatment, and samples from aerial regions of seedlings from control seeds (green line) and fengycin-treated seeds (purple line) five days after treatment. Orange line represents the same feature in samples from a *B. subtilis* NCIB 3610 colony, used as positive control.

5) When testing the interaction of TasA and fengycin with the seed oil bodies, *B. subtilis* produced other SMs should also be included, as proposed, fengycin can induce the formation of the pores and subsequent changes in membrane permeability, *B. subtilis* produced surfactins also showed this function in previous reports of this strain. Currently, the functional mechanism of fengycin is mostly speculated.

R- The combination of the findings on promotion activity after treatments with cells of diverse mutants in the ECM, or purified molecules consistently indicated fengycin as the most prominent molecule inducing the growth of the plant, and thus all the study was focused on this relevant molecule. A mutant in surfactin still retained promotion activity, and we did not observe a promotion effect with pure surfactin. However, as suggested by the reviewer, we have done new experiments, and have found that surfactin leads to a similar disgregation of purified OBs and also *in vivo* OBs (Figure R2a,c).

Additionally, our metabolomic analysis highlights an increase of GSH after fengycin treatment indicating a possible implication of ROS accumulation. New experiments in this direction have revealed that fengycin induces the production of ROS after seed imbibition (see new Figure 4), however the levels are much higher than those recorded after the treatment with surfactin (Figure R2b,d). We propose that the balance between ROS acting as a beneficial stimulus and specific metabolic activity triggered after oil bodies degradation provokes the accumulation of certain families of lipids related molecules and glutathione which together contribute to the stimulatory effect.

As the reviewer suggests, surfactin also targets membranes, however, it is also well known the differential antimicrobial activity of these molecules, so it is not surprising that the levels of ROS generated after surfactin treatment are different from those related to fengycin treatment. Thus, we do not disregard that surfactin, or other lipopeptides may contribute at some point to the overall effect of *Bacillus* cells, however, a comparative study of the stimulatory effect of diverse lipopeptides (LPs) is a

more complex study that should take also into consideration the physicochemical properties and biological impact of these LPs in a concentration dependent manner, which is beyond the scope of this manuscript. This is a biological question that we are exploring in our laboratory as a part of future work.

We have, however, rewritten the results, to focus on the fengycin effect, with no mention of other LPs, as surfactin, in order to avoid misinterpretations.Figure R2

Figure R2. a Representative CLSM images of purified OBs suspension (up) or transversely cut seeds (down) 16 hours after the treatment with water (control), fengycin or surfactin. Oil bodies were stained with Nile red. **b** Representative CLSM images of the inner tissues of Dihydrorhodamine-stained seeds immediately after seed treatments with water (control), fengycin or surfactin. **c** Plot representing the OB areas 16 hours after the treatment of a purified OBs suspension with water (control), fengycin or surfactin, measured as the area of particles from three different fields. Average values are shown, and error bars represent SD. Statistical significance was assessed by one-way ANOVA with multiple comparisons test. **d** Plot representing the intensity of the fluorescence in seeds immediately after treatment with water, fengycin or surfactin, measured as the mean gray value from n=30 random sections from three different fields. Average values are shown, and error bars represent SD. Statistical significance was assessed by one-way ANOVA with multiple comparisons test.

6) line 173-176, these conclusions are overstated, supporting evidences are not enough.

R- Thank you. We have toned down these conclusions. See lines 175-179.

7) line 264, Fig. 1F should be Fig. 3F.

R- Thank you.

Decision Letter, first revision:

Dear Professor Romero,

Thank you for your patience while your manuscript "The crosstalk between Bacillus extracellular matrix and seed lipid storages stimulates overgrowth and immunization of plants" was under peer-review at Nature Microbiology. It has now been seen by 3 referees, whose expertise and comments you will find at the of this email. You will see from their comments below that while two reviewers are satisfied, the third reviewer has made detailed and specific requests about presentation of the metabolomics data that we feel are required before we can proceed.

In summary, we are very interested in the possibility of publishing your study in Nature Microbiology, but would like to consider your response to these concerns in the form of a revised manuscript before we make a final decision on publication.

If you have not done so already please begin to revise your manuscript so that it conforms to our Article format instructions at <http://www.nature.com/nmicrobiol/info/final-submission/>

15The usual length limit for a Nature Microbiology Article is six display items (figures or tables) and 3,000 words. We have some flexibility, and can allow a revised manuscript at 3,500 words, but please consider this a firm upper limit. There is a trade-off of ~250 words per display item, so if you need more space, you could move a Figure or Table to Supplementary Information.

Some reduction could be achieved by focusing any introductory material and moving it to the start of your opening 'bold' paragraph, whose function is to outline the background to your work, describe in a sentence your new observations, and explain your main conclusions. The discussion should also be limited. Methods should be described in a separate section following the discussion, we do not place a word limit on Methods.

Nature Microbiology titles should give a sense of the main new findings of a manuscript, and should not contain punctuation. Please keep in mind that we strongly discourage active verbs in titles, and that they should ideally fit within 90 characters each (including spaces).

Please include a data availability statement as a separate section after Methods but before references, under the heading "Data Availability". This section should inform readers about the availability of the data used to support the conclusions of your study. This information includes accession codes to public repositories (data banks for protein, DNA or RNA sequences, microarray, proteomics data etc...), references to source data published alongside the paper, unique identifiers such as URLs to data repository entries, or data set DOIs, and any other statement about data availability. At a minimum, you should include the following statement: "The data that support the findings of this study are available from the corresponding author upon request", mentioning any restrictions on availability. If DOIs are provided, we also strongly encourage including these in the Reference list (authors, title, publisher (repository name), identifier, year). For more guidance on how to write this section please see:

<http://www.nature.com/authors/policies/data/data-availability-statements-data-citations.pdf>

To improve the accessibility of your paper to readers from other research areas, please pay particular attention to the wording of the paper's opening bold paragraph, which serves both as an introduction and as a brief, non-technical summary in about 150 words. If, however, you require one or two extra sentences to explain your work clearly, please include them even if the paragraph is over-length as a result. The opening paragraph should not contain references. Because scientists from other sub-disciplines will be interested in your results and their implications, it is important to explain essential but specialised terms concisely. We suggest you show your summary paragraph to colleagues in other fields to uncover any problematic concepts.

16If your paper is accepted for publication, we will edit your display items electronically so they conform to our house style and will reproduce clearly in print. If necessary, we will re-size figures to fit single or double column width. If your figures contain several parts, the parts should form a neat rectangle when assembled. Choosing the right electronic format at this stage will speed up the processing of your paper and give the best possible results in print. We would like the figures to be supplied as vector files - EPS, PDF, AI or postscript (PS) file formats (not raster or bitmap files), preferably generated with vector-graphics software (Adobe Illustrator for example). Please try to ensure that all figures are non-flattened and fully editable. All images should be at least 300 dpi resolution (when figures are scaled to approximately the size that they are to be printed at) and in RGB colour format. Please do not submit Jpeg or flattened TIFF files. Please see also 'Guidelines for Electronic Submission of Figures' at the end of this letter for further detail.

Figure legends must provide a brief description of the figure and the symbols used, within 350 words, including definitions of any error bars employed in the figures.

Please include a statement before the acknowledgements naming the author to whom correspondence and requests for materials should be addressed.

Finally, we require authors to include a statement of their individual contributions to the paper -- such as experimental work, project planning, data analysis, etc. -- immediately after the acknowledgements. The statement should be short, and refer to authors by their initials. For details please see the Authorship section of our joint Editorial policies at http://www.nature.com/authors/editorial_policies/authorship.html

* include a point-by-point response to any editorial suggestions and to our referees. Please include your response to the editorial suggestions in your cover letter, and please upload your response to the

17referees as a separate document.

* ensure it complies with our format requirements for Letters as set out in our guide to authors at www.nature.com/nmicrobiol/info/gta/

* state in a cover note the length of the text, methods and legends; the number of references; number and estimated final size of figures and tables

* resubmit electronically if possible using the link below to access your home page:

{redacted}

*This url links to your confidential homepage and associated information about manuscripts you may have submitted or be reviewing for us. If you wish to forward this e-mail to co-authors, please delete this link to your homepage first.

Please ensure that all correspondence is marked with your Nature Microbiology reference number in the subject line.

Nature Microbiology is committed to improving transparency in authorship. As part of our efforts in this direction, we are now requesting that all authors identified as 'corresponding author' on published papers create and link their Open Researcher and Contributor Identifier (ORCID) with their account on the Manuscript Tracking System (MTS), prior to acceptance. This applies to primary research papers only. ORCID helps the scientific community achieve unambiguous attribution of all scholarly contributions. You can create and link your ORCID from the home page of the MTS by clicking on 'Modify my Springer Nature account'. For more information please visit please visit www.springernature.com/orcid.

We hope to receive your revised paper within three weeks. If you cannot send it within this time, please let us know.

Yours sincerely,

{redacted}

Reviewers Comments:

Reviewer #1 (Remarks to the Author):

In the revised manuscript, authors added additional experiments to address the concerns from the

18previous reviewers, and also tried to explain some criticisms, accordingly, the manuscript was carefully revised and significantly improved. Although I still have reservations for partial previous comments, I don't have new suggestions, and support the acceptance of the revised manuscript.

Reviewer #2 (Remarks to the Author):

This article is a deeply revised version of a manuscript which was submitted a couple months ago. The authors have performed new experiments and additional analysis, answering most of my comments; other unanswered questions from my side were met with satisfactory explanations. The story is now much tighter and conclusions are well supported. This is a great article.

Reviewer #3 (Remarks to the Author):

I was largely positive about this manuscript in the first round of review, and I am still quite enthusiastic about these findings. However, the metabolomics aspects of this manuscript still require attention as they are not yet up to field standards.

Major concerns

1: Issues remain with compound identification.

I appreciate that the authors have now used the Sumner et al 2007 framework to categorize most of the putative metabolites that they mention throughout the paper. Unfortunately, the characterization of small molecules here is still falling short of normal standards in this field. Specifically, what is lacking is a table that summarizes these chemical features, and their basis for being categorized as identified as level 2 within the Sumner et al 2007 framework. At a minimum I would look for this table to include: Ion mode, observed m/z, calculated m/z, ppm error, proposed compound name, and level of ID. My expectation for a level 2 ID is that it should have at least two orthogonal data types that back up this ID. This could be the precursor m/z within ~5 ppm error and an MS2 spectrum that matches a known compound in a database like GNPS. A typical example of such reporting can be seen in supplemental table S7 from this paper:

Combined transcriptomic and metabolomic analysis reveals the potential mechanism of seed germination and young seedling growth in *Tamarix hispida*.

Pang X, Suo J, Liu S, Xu J, Yang T, Xiang N, Wu Y, Lu B, Qin R, Liu H, Yao J.
BMC Genomics. 2022 Feb 8;23(1):109. doi: 10.1186/s12864-022-08341-x.

I would expect that any named compound will also have clearly labeled mirror plots of MS2 spectra in the supplemental info. That would include Mono/Dilinolenin, Fengycin, L-tryptophan, cinnamic acid, etc. Beyond this, the glutathione analogs identified as level 2 in Fig 4B appear different in 6D (m/z 308 vs 311). What accounts for this? I will also point out that right now the two mirror plots provided lack any identifying information save for the m/z.

19Minor concerns:

Abstract: "We propose this mutualistic interaction is conserved in Bacilli and plant seeds containing storage oil bodies." What is the evidence for this? In order to make such a claim, we would need a functional analysis across multiple species of bacillus and multiple types of seeds. This is an evolutionary claim, and would need to further be corroborated with experiments that take evolution into account. I think it would be better to just settle on a different final summary statement.

Figure 1e: What is the scale bar in the upper right? The numbers here are too small to read, and the color labels are also unclear.

Figure 4a: Please state concentration/amount of fengycin, TasA, BslA, and surfactin used for treatment in the pictures of radicles.

Author Rebuttal, first revision:

Reviewer #1:

In the revised manuscript, authors added additional experiments to address the concerns from the previous reviewers, and also tried to explain some criticisms, accordingly, the manuscript was carefully revised and significantly improved. Although I still have reservations for partial previous comments, I don't have new suggestions, and support the acceptance of the revised manuscript.

R- Thank you for your kind words and dedication for the improvement of the manuscript.

Reviewer #2 :

This article is a deeply revised version of a manuscript which was submitted a couple months ago. The authors have performed new experiments and additional analysis, answering most of my comments; other unanswered questions from my side were met with satisfactory explanations. The story is now much tighter and conclusions are well supported. This is a great article.

R- We really appreciate your kind words and dedication for the improvement of the manuscript.

20Reviewer #3

I was largely positive about this manuscript in the first round of review, and I am still quite enthusiastic about these findings. However, the metabolomics aspects of this manuscript still require attention as they are not yet up to field standards.

R- Thank you for your kind words and dedication for the improvement of the manuscript.

Major concerns:

1: Issues remain with compound identification.

I appreciate that the authors have now used the Sumner et al 2007 framework to categorize most of the putative metabolites that they mention throughout the paper. Unfortunately, the characterization of small molecules here is still falling short of normal standards in this field. Specifically, what is lacking is a table that summarizes these chemical features, and their basis for being categorized as identified as level 2 within the Sumner et al 2007 framework. At a minimum I would look for this table to include: Ion mode, observed m/z, calculated m/z, ppm error, proposed compound name, and level of ID. My expectation for a level 2 ID is that it should have at least two orthogonal data types that back up this ID. This could be the precursor m/z within ~5 ppm error and an MS2 spectrum that matches a known compound in a database like GNPS. A typical example of such reporting can be seen in supplemental table S7 from this paper:

Combined transcriptomic and metabolomic analysis reveals the potential mechanism of seed germination and young seedling growth in Tamarix hispida.
Pang X, Suo J, Liu S, Xu J, Yang T, Xiang N, Wu Y, Lu B, Qin R, Liu H, Yao J.
BMC Genomics. 2022 Feb 8;23(1):109. doi: 10.1186/s12864-022-08341-x.

R- We have added a new supplementary table (Table S8) with the information requested.

I would expect that any named compound will also have clearly labeled mirror plots of MS2 spectra in the supplemental info. That would include Mono/Dilinolenin, Fengycin, L-tryptophan, cinnamic acid, etc.

21R- We have included the mirror plots of any named metabolite along the manuscript in Figure S12.

Beyond this, the glutathione analogs identified as level 2 in Fig 4B appear different in 6D (m/z 308 vs 311). What accounts for this?

R- We understand that the reviewer is referring to m/z 611 instead 311 in Fig. 6D (now 5D). The differences are due to the oxidation status of the metabolite: in Fig 4B (now 3B) we found glutathione reduced and in Fig 6D (now 5D) the glutathione appears oxidized, with a higher m/z.

I will also point out that right now the two mirror plots provided lack any identifying information save for the m/z.

R- Thank you for the comment. Updated mirror plots in Fig. S12 now contains identifying information besides m/z such as delta m/z, cosine and name of the compound in the library.

Minor concerns:

Abstract: "We propose this mutualistic interaction is conserved in Bacilli and plant seeds containing storage oil bodies." What is the evidence for this? In order to make such a claim, we would need a functional analysis across multiple species of bacillus and multiple types of seeds. This is an evolutionary claim, and would need to further be corroborated with experiments that take evolution into account. I think it would be better to just settle on a different final summary statement.

R- The reviewer is totally right. We have considered this concern and decided to write a new final statement without referring to evolutionary topics.

Figure 1e: What is the scale bar in the upper right? The numbers here are too small to read, and the color labels are also unclear.

R- The scale bar in the upper right represents the contribution of each metabolite from the top 100 metabolites represented (rectangles inside the bar colored in function to their chemical class) to the

22total abundance (100%), ordered from highest to lowest contribution. We have updated the figure legend to clarify this information.

In relation to the figure, the reviewer is totally right and we have increased the size of the scale bar and the font of the percentages.

Figure 4a: Please state concentration/amount of fengycin, TasA, BslA, and surfactin used for treatment in the pictures of radicles.

R- We have included this information (see Fig. 3a).

Decision Letter, second revision:

Dear Professor Romero,

Thank you for your patience while your manuscript "Bacillus subtilis extracellular matrix stimulates the growth and immunization of plants by targeting the seed storages" was under peer-review at Nature Microbiology. It has now been seen by one of the previous referees, who has now laid out very clearly the community expectations for presentation of some of your data.

Please proceed to address this concern and resubmit a final revision. As long as this is done to our satisfaction we do not anticipate further peer review.

If you have not done so already please begin to revise your manuscript so that it conforms to our Article format instructions at <http://www.nature.com/nmicrobiol/info/final-submission/>

The usual length limit for a Nature Microbiology Article is six display items (figures or tables) and 3,000 words. We have some flexibility, and can allow a revised manuscript at 3,500 words, but please consider this a firm upper limit. There is a trade-off of ~250 words per display item, so if you need more space, you could move a Figure or Table to Supplementary Information.

Some reduction could be achieved by focusing any introductory material and moving it to the start of your opening 'bold' paragraph, whose function is to outline the background to your work, describe in a sentence your new observations, and explain your main conclusions. The discussion should also be limited. Methods should be described in a separate section following the discussion, we do not place a word limit on Methods.

23Nature Microbiology titles should give a sense of the main new findings of a manuscript, and should not contain punctuation. Please keep in mind that we strongly discourage active verbs in titles, and that they should ideally fit within 90 characters each (including spaces).

Please include a data availability statement as a separate section after Methods but before references, under the heading "Data Availability". This section should inform readers about the availability of the data used to support the conclusions of your study. This information includes accession codes to public repositories (data banks for protein, DNA or RNA sequences, microarray, proteomics data etc...), references to source data published alongside the paper, unique identifiers such as URLs to data repository entries, or data set DOIs, and any other statement about data availability. At a minimum, you should include the following statement: "The data that support the findings of this study are available from the corresponding author upon request", mentioning any restrictions on availability. If DOIs are provided, we also strongly encourage including these in the Reference list (authors, title, publisher (repository name), identifier, year). For more guidance on how to write this section please see:

<http://www.nature.com/authors/policies/data/data-availability-statements-data-citations.pdf>

To improve the accessibility of your paper to readers from other research areas, please pay particular attention to the wording of the paper's opening bold paragraph, which serves both as an introduction and as a brief, non-technical summary in about 150 words. If, however, you require one or two extra sentences to explain your work clearly, please include them even if the paragraph is over-length as a result. The opening paragraph should not contain references. Because scientists from other sub-disciplines will be interested in your results and their implications, it is important to explain essential but specialised terms concisely. We suggest you show your summary paragraph to colleagues in other fields to uncover any problematic concepts.

If your paper is accepted for publication, we will edit your display items electronically so they conform to our house style and will reproduce clearly in print. If necessary, we will re-size figures to fit single or double column width. If your figures contain several parts, the parts should form a neat rectangle when assembled. Choosing the right electronic format at this stage will speed up the processing of your paper and give the best possible results in print. We would like the figures to be supplied as vector files - EPS, PDF, AI or postscript (PS) file formats (not raster or bitmap files), preferably generated with vector-graphics software (Adobe Illustrator for example). Please try to ensure that all figures are non-flattened and fully editable. All images should be at least 300 dpi resolution (when figures are scaled to approximately the size that they are to be printed at) and in RGB colour format. Please do not submit Jpeg or flattened TIFF files. Please see also 'Guidelines for Electronic Submission of Figures' at the end of this letter for further detail.

24Figure legends must provide a brief description of the figure and the symbols used, within 350 words, including definitions of any error bars employed in the figures.

Please ensure that you retain unprocessed data and metadata files after publication, ideally archiving data in perpetuity, as these may be requested during the peer review and production process or after publication if any issues arise.

Please include a statement before the acknowledgements naming the author to whom correspondence and requests for materials should be addressed.

Finally, we require authors to include a statement of their individual contributions to the paper -- such as experimental work, project planning, data analysis, etc. -- immediately after the acknowledgements. The statement should be short, and refer to authors by their initials. For details please see the Authorship section of our joint Editorial policies at http://www.nature.com/authors/editorial_policies/authorship.html

- * include a point-by-point response to any editorial suggestions and to our referees. Please include your response to the editorial suggestions in your cover letter, and please upload your response to the referees as a separate document.
- * ensure it complies with our format requirements for Letters as set out in our guide to authors at www.nature.com/nmicrobiol/info/gta/
- * state in a cover note the length of the text, methods and legends; the number of references; number and estimated final size of figures and tables
- * resubmit electronically if possible using the link below to access your home page:

{redacted}

*This url links to your confidential homepage and associated information about manuscripts you may have submitted or be reviewing for us. If you wish to forward this e-mail to co-authors, please delete this link to your homepage first.

Please ensure that all correspondence is marked with your Nature Microbiology reference number in the subject line.

Nature Microbiology is committed to improving transparency in authorship. As part of our efforts in this direction, we are now requesting that all authors identified as 'corresponding author' on published papers create and link their Open Researcher and Contributor Identifier (ORCID) with their account on the Manuscript Tracking System (MTS), prior to acceptance. This applies to primary research papers only. ORCID helps the scientific community achieve unambiguous attribution of all scholarly contributions. You can create and link your ORCID from the home page of the MTS by clicking on 'Modify my Springer Nature account'. For more information please visit www.springernature.com/orcid.

We hope to receive your revised paper within three weeks. If you cannot send it within this time, please let us know.

{redacted}

Reviewers Comments:

Reviewer #3 (Remarks to the Author):

I thank the authors for their continued improvements to this manuscript. I think it is now ready for publication, except for providing one table in supplemental regarding the metabolomic identifications. This table should include, for all compounds identified in Figure S12:

1. proposed compound name,
2. Ion mode
3. Adduct
4. observed m/z
5. calculated m/z
6. ppm error
7. level of ID

I realize that this may seem superfluous to authors. However, it is not. This table provides the justification of the identification of the different compounds named by the authors, and I would consider it a minimal bar to clear. It is important for several reasons: 1. This is the standard of the

26field, 2. Your audience needs a clear understanding of your confidence in these identifications, and 3. Tables like this guard against spurious identifications (and the implied chemical connections) in untargeted metabolomics datasets. A paper of this magnitude will come with great interest and intense scrutiny. Adhering to field standards now will benefit the authors later.

The data for the requested table will need to be curated by the authors. The 77 pages of tables recently provided as supplemental is basically raw analytical output. It is not helpful since there is no way to correlate data from the first set of tables with the second. These data should be removed as they are not helpful toward the goal of understanding the identifications made by the authors.

Author Rebuttal, second revision:

Reviewer #3:

I thank the authors for their continued improvements to this manuscript. I think it is now ready for publication, except for providing one table in supplemental regarding the metabolomic identifications. This table should include, for all compounds identified in Figure S12:

- 1. proposed compound name,**
- 2. Ion mode**
- 3. Adduct**
- 4. observed m/z**
- 5. calculated m/z**
- 6. ppm error**
- 7. level of ID**

I realize that this may seem superfluous to authors. However, it is not. This table provides the justification of the identification of the different compounds named by the authors, and I would consider it a minimal bar to clear. It is important for several reasons: 1. This is the standard of the field, 2. Your audience needs a clear understanding of your confidence in these identifications, and 3. Tables like this guard against spurious identifications (and the implied chemical connections) in untargeted metabolomics datasets. A paper of this magnitude will come with great interest and intense scrutiny. Adhering to field standards now will benefit the authors later.

The data for the requested table will need to be curated by the authors. The 77 pages of tables recently provided as supplemental is basically raw analytical output. It is not helpful since there is no way to correlate data from the first set of tables with the second.

27These data should be removed as they are not helpful toward the goal of understanding the identifications made by the authors.

R – We really appreciate reviewer comments. We agree with the reviewer view on the importance of clarity and justification of all the metabolites mentioned in the manuscript, specifically those highlighted in Fig. S12. Therefore, we have included a new table (see Table S8) with the information requested. As suggested, we have deleted the previous table (old Table S8) in order to avoid confusion in the understanding of the data.

Decision Letter, third revision:

Dear Dr. Romero,

Thank you for submitting your revised manuscript "Bacillus subtilis extracellular matrix stimulates the growth and immunization of plants by targeting the seed storages" (NMICROBIOL-21092279C). With the addition of Table S8 in line with reviewers guidance, we are now happy in principle to publish it in Nature Microbiology, pending minor revisions to comply with our editorial and formatting guidelines.

Thank you again for your interest in Nature Microbiology Please do not hesitate to contact me if you have any questions.
{redacted}

Decision Letter, final checks

Dear Dr. Romero,

Thank you for your patience as we've prepared the guidelines for final submission of your Nature Microbiology manuscript, "Bacillus subtilis extracellular matrix stimulates the growth and immunization of plants by targeting the seed storages" (NMICROBIOL-21092279C). I'm sorry that this took a little longer than I would have liked.

Please carefully follow the step-by-step instructions provided in the attached file, and add a response

28in each row of the table to indicate the changes that you have made.

Please also check and comment on any additional marked-up edits we have proposed within the attached and edited text. Ensuring that each point is addressed will help to ensure that your revised manuscript can be swiftly handed over to our production team.

In recognition of the time and expertise our reviewers provide to Nature Microbiology's editorial process, we would like to formally acknowledge their contribution to the external peer review of your manuscript entitled "Bacillus subtilis extracellular matrix stimulates the growth and immunization of plants by targeting the seed storages". For those reviewers who give their assent, we will be publishing their names alongside the published article.

Nature Microbiology offers a Transparent Peer Review option for new original research manuscripts submitted after December 1st, 2019. As part of this initiative, we encourage our authors to support increased transparency into the peer review process by agreeing to have the reviewer comments, author rebuttal letters, and editorial decision letters published as a Supplementary item. When you submit your final files please clearly state in your cover letter whether or not you would like to participate in this initiative. Please note that failure to state your preference will result in delays in accepting your manuscript for publication.

Cover suggestions

As you prepare your final files we encourage you to consider whether you have any images or illustrations that may be appropriate for use on the cover of Nature Microbiology.

29Please submit your suggestions, clearly labeled, along with your final files. We'll be in touch if more information is needed.

Nature Microbiology has now transitioned to a unified Rights Collection system which will allow our Author Services team to quickly and easily collect the rights and permissions required to publish your work. Approximately 10 days after your paper is formally accepted, you will receive an email in providing you with a link to complete the grant of rights. If your paper is eligible for Open Access, our Author Services team will also be in touch regarding any additional information that may be required to arrange payment for your article.

Please note that *Nature Microbiology* is a Transformative Journal (TJ). Authors may publish their research with us through the traditional subscription access route or make their paper immediately open access through payment of an article-processing charge (APC). Authors will not be required to make a final decision about access to their article until it has been accepted. [Find out more about Transformative Journals](https://www.springernature.com/gp/open-research/transformative-journals)

Please use the following link for uploading these materials:
{redacted}

Best regards,

{redacted}

Final Decision Letter:

25th April 2022

Dear Professor Romero,

I am pleased to accept your Article "Bacillus subtilis biofilm matrix components target seed oil bodies to promote growth and anti-fungal resistance in melon" for publication in Nature Microbiology. Thank you for having chosen to submit your work to us and many congratulations.

Acceptance of your manuscript is conditional on all authors' agreement with our publication policies (see <https://www.nature.com/nmicrobiol/editorial-policies>). In particular your manuscript must not be published elsewhere and there must be no announcement of the work to any media outlet until the publication date (the day on which it is uploaded onto our website).

Please note that *Nature Microbiology* is a Transformative Journal (TJ). Authors may publish their research with us through the traditional subscription access route or make their paper immediately open access through payment of an article-processing charge (APC). Authors will not be required to make a final decision about access to their article until it has been accepted. [Find out more about Transformative Journals](https://www.springernature.com/gp/open-research/transformative-journals)

Authors may need to take specific actions to achieve [a](https://www.springernature.com/gp/open-research/funding/policy-compliance-)

31FAQs" > compliance with funder and institutional open access mandates. If your research is supported by a funder that requires immediate open access (e.g. according to Plan S principles) then you should select the gold OA route, and we will direct you to the compliant route where possible. For authors selecting the subscription publication route, the journal's standard licensing terms will need to be accepted, including self-archiving policies. Those licensing terms will supersede any other terms that the author or any third party may assert apply to any version of the manuscript.

An online order form for reprints of your paper is available at https://www.nature.com/reprints/author-reprints.html. All co-authors, authors' institutions and authors' funding agencies can order reprints using the form appropriate to their geographical region.

With kind regards,

{redacted}